# Impaired DNA damage response signaling by FUS-NLS mutations leads to neurodegeneration and FUS aggregate formation

Marcel Naumann[1], Arun Pal[1], Anand Goswami[2], Xenia Lojewski[1], Julia Japtok[1], Anne Vehlow[3,4,5], Maximilian Naujock[6,21], René Günther[1], Mengmeng Jin[1], Nancy Stanslowsky[6], Peter Reinhardt[7,22], Jared Sterneckert[7], Marie Frickenhaus[8,9], Francisco Pan-Montojo[10], Erik Storkebaum[8,9,23], Ina Poser[11], Axel Freischmidt[12], Jochen H. Weishaupt[12], Karlheinz Holzmann[13], Dirk Troost[14], Albert C. Ludolph[12], Tobias M. Boeckers[15], Stefan Liebau[16], Susanne Petri[6], Nils Cordes[3,4,5,17], Anthony A. Hyman[11], Florian Wegner[6], Stephan W. Grill[11,18], Joachim Weis[2], Alexander Storch[1,7,19,20] & Andreas Hermann[1,7,19]

Amyotrophic lateral sclerosis (ALS) is the most frequent motor neuron disease. Cytoplasmic fused in sarcoma (FUS) aggregates are pathological hallmarks of FUS-ALS. Proper shuttling between the nucleus and cytoplasm is essential for physiological cell function. However, the initial event in the pathophysiology of FUS-ALS remains enigmatic. Using human induced pluripotent stem cell (hiPSCs)-derived motor neurons (MNs), we show that impairment of poly(ADP-ribose) polymerase (PARP)-dependent DNA damage response (DDR) signaling due to mutations in the FUS nuclear localization sequence (NLS) induces additional cyto-plasmic FUS mislocalization which in turn results in neurodegeneration and FUS aggregate formation. Our work suggests that a key pathophysiologic event in ALS is upstream of aggregate formation. Targeting DDR signaling could lead to novel therapeutic routes for ameliorating ALS.

[1] Department of Neurology, Technische Universität Dresden, 01307 Dresden, Germany. [2] Institute of Neuropathology, RWTH Aachen University Hospital, Pauwelsstrasse-30, 52074 Aachen, Germany. [3] OncoRay—National Center for Radiation Research in Oncology, Faculty of Medicine and University Hospital Carl Gustav Carus, Technische Universität Dresden, Helmholtz-Zentrum Dresden-Rossendorf, Dresden 01307, Germany. [4] German Cancer Consortium (DKTK), partner site Dresden, and German Cancer Research Center (DKFZ), 69192 Heidelberg, Germany. [5] Helmholtz-Zentrum Dresden-Rossendorf, Institute of Radiooncology—OncoRay, 01328 Dresden, Germany. [6] Department of Neurology, Hannover Medical School, 30625 Hannover, Germany. [7] Center for Regenerative Therapies Dresden (CRTD), Technische Universität Dresden, 01307 Dresden, Germany. [8] Molecular Neurogenetics Laboratory, Max Planck Institute for Molecular Biomedicine, 48149 Münster, Germany. [9] Faculty of Medicine, University of Münster, 48149 Münster, Germany. [10] Department of Neurology, Klinikum der Universität München, and Munich Cluster for Systems Neurology, SyNergy, 81377 Munich, Germany. [11] Max Planck Institute of Molecular Cell Biology and Genetics, 01307 Dresden, Germany. [12] Department of Neurology, University Ulm, 89081 Ulm, Germany. [13] Genomics-Core Facility, University Hospital Ulm, Centre for Biomedical Research, 89081 Ulm, Germany. [14] Division of Neuropathology, Department of Pathology, Academic Medical Centre, 1105 AZ Amsterdam, The Netherlands. [15] Institute of Anatomy and Cell Biology, University of Ulm, 89081 Ulm, Germany. [16] Institute of Neuroanatomy & Developmental Biology, Eberhard Karls University of Tübingen, 72074 Tübingen, Germany. [17] Department of Radiotherapy and Radiation Oncology, Faculty of Medicine and University Hospital Carl Gustav Carus, Technische Universität Dresden, 01307 Dresden, Germany. [18] BIOTEC, Technische Universität Dresden, 01307 Dresden, Germany. [19] German Center for Neurodegenerative Diseases (DZNE), Research Site Rostock, 18147 Rostock, Germany. [20] Department of Neurology, University of Rostock, 18147 Rostock, Germany. [21] Present address: CNS Research Department, Boehringer Ingelheim Pharma GmbH & Co. KG, Binger Strasse 173, 55216 Ingelheim am Rhein, Germany. [22] Present address: AbbVie Deutschland GmbH & Co. KG, 67061 Ludwigshafen, Germany. [23] Present address: Department of Molecular Neurobiology, Donders Institute for Brain, Cognition and Behaviour, Radboud University, Nijmegen, The Netherlands. Marcel Naumann and Arun Pal contributed equally to this work. Correspondence and requests for materials should be addressed to A.H. (email: Andreas.Hermann@uniklinikum-dresden.de)

Amyotrophic lateral sclerosis (ALS) is a devastating neurodegenerative disease leading to death within 2–5 years of symptom onset. Fused in Sarcoma (FUS) is one of the most frequently mutated genes in familial ALS (fALS), being responsible for approx. 5% of fALS and up to 1% of sporadic ALS (sALS)[1,2] cases. Autosomal-dominant mutations within the nuclear localization signal (NLS) region of FUS are by far the most prevalent mutations and clearly pathogenic[3], with the R521C and R521H point mutations being the most common[2]. While physiological FUS function depends on proper shuttling between the nucleus and cytoplasm, cytoplasmic FUS aggregates are a pathological hallmark of FUS-ALS.

FUS mislocalization due to nucleo-cytoplasmic shuttling[4] depends on two main pathways. First, Transportin (TRN)-mediated nuclear import of FUS is known to be disrupted by FUS-NLS mutations[4–6]. Arginine methylation of the PY-NLS domain modulates TRN binding to FUS and its nuclear import. Inhibition of arginine methylation is known to restore TRN-mediated nuclear import in FUS-NLS mutant HeLa cell culture models[5]. Similarly, FUS+ aggregates in ALS postmortem specimens contain methylated FUS[5], which was also recently reported for iPSC-derived cortical neurons[7]. Second, Deng and colleagues reported DNA-damage-induced FUS phosphorylation by the DNA-dependent protein kinase (DNA-PK), leading to nuclear export of FUS[8].

Previous reports on human motor neuronal cell culture models of FUS-ALS showed the acquirement of typical neuropathology, such as cytoplasmic mislocalization of mutant FUS as well as appearance of FUS+ cytoplasmic inclusions[7,9–11]. However, mechanistic insights into how these events cause neurodegeneration and about upstream events are still lacking.

FUS is physiologically involved in RNA metabolism (transcription, splicing, and export to cytoplasm) and DNA repair[3]. Recent data suggest a significant role in DNA damage response (DDR) downstream of poly(ADP-ribose) polymerase (PARP) not involving ATM or DNA-dependent protein kinase (DNA-PK)[12–14]. DNA damage is the primary activator of PAR polymerase 1 (PARP1) that catalyzes the reaction of poly(ADP-ribosylation) (for review see ref. [15]). Previous studies showed that FUS is rapidly recruited to DNA damage sites (DDS) in a PAR-dependent manner[13,14,16]. Indeed, PARP1 arrives within seconds of DNA damage followed immediately by FUS[17]. PAR is degraded by poly(ADP-ribose) glycohydrolase (PARG)[18] and PARG inhibition leads to prolonged recruitment of FUS to DDS[17]. In addition, an interaction of FUS and Histone deacetylase 1 (HDAC1) was reported to be diminished by FUS NLS mutations resulting in impairment of proper DDR[14,19]. FUS directly interacts with PAR[13] and PARylation was shown to induce additional PARP1 recruitment to DDS[20].

Wang and colleagues reported two FUS-NLS cases that exhibited increased DNA damage in the postmortem motor cortex[14]. In addition, increased levels of oxidative DNA damage were reported in the spinal cord of both sporadic and familial ALS patients[21]. While mice carrying FUS NLS mutations also showed signs of increased DNA damage[19], FUS−/− mice have obvious signs of genetic instability[22].

Recent studies suggest that PARP is involved in forming liquid compartments of FUS at DDS, and that aberrant phase transition of the liquid compartments to solid-like aggregates could be involved in the onset of the disease[17,23–25]. However, the relationship between DNA damage and the formation of cytoplasmic aggregates and to neurodegeneration is unknown. Here, we (i) develop a human MN model of FUS-ALS with endogenously tagged protein, (ii) investigate DNA damage in MNs and (iii) link DDR signaling to aggregate formation and neurodegeneration. Moreover, we report a neuronopathy with distal axon degeneration as the major phenotype of FUS-ALS prior to FUS aggregation. Furthermore, we show that inappropriate DDR signaling due to FUS NLS mutations is a key upstream event in FUS-ALS enhancing/inducing a vicious cycle by increasing cytoplasmic FUS shuttling. This study suggests that targeting DNA damage could be a new therapeutic strategy for ALS.

## Results

**Patient-specific FUSmt motor neurons reproduce key pathology.** To develop a human MN model of FUS-ALS, we generated human induced pluripotent stem cells (hiPSCs), by classical retroviral "Yamanaka-factor" reprogramming, from three different FUS-ALS patients carrying diverse NLS mutations (R521C, R521L, R495QfsX527; Fig. 1, Table 1). Additionally, we generated isogenic iPSC lines by CRISPR/Cas9n from one clone of the R521C hiPSC lines by generating both a wildtype and a new (P525L) mutation carrying an additional c-terminal GFP tag (Supplementary Fig. 1). We included only fully characterized hiPSC with a normal karyotype and confirmed mutations in our study (see Methods). We generated fully functional MNs and then tested for acquisition of hallmark pathology (Fig. 1a–e)[26]. Spinal MN differentiation yielded ≈50% MNs with no difference between healthy controls and FUS-ALS patient lines (Fig. 1c). MNs expressed typical markers for spinal MNs, including HB9, Islet, SMI32 and ChAT (Fig. 1b, e)[27]. We neither observed increased cell death nor pathological FUS aggregation in the early stages of MN differentiation (Fig. 1c, f). Electrophysiology revealed the presence of voltage-gated sodium and potassium channels (Fig. 1g, h, l), firing of evoked and spontaneous action potentials (Fig. 1i, j) and periodical spontaneous increases of intracellular calcium (Fig. 1k), providing evidence of neuronal function[28]. Interestingly, we show a hypoexcitability in FUSmt MNs that was abolished by genotype correction (Fig. 1m–p). After extended maturation (>30 days of differentiation), FUS-ALS—but not control-derived—spinal MNs increasingly showed cytoplasmic FUS translocation and spontaneous appearance of cytoplasmic FUS inclusions (Fig. 1f). Those inclusions were also positive for methylated FUS as typically seen in FUS-ALS (Fig. 1f)[5,7]. Immunoblot analysis of spinal MNs further confirmed Triton-x insoluble FUS aggregates (dot blot) and increased polyubiquitinylation in R521C FUS MNs (Fig. 1d, Supplementary Fig. 11). Taken together, we have developed an iPSC-based human spinal MN disease model of FUS-ALS showing normal differentiation into fully functional spinal MNs with subsequent acquisition of hallmark pathology—including neuronal dysfunction and protein aggregation—during cellular aging. This model is ideal for pathophysiological studies.

**Mutant FUS predominantly affects distal axons.** To further characterize the FUSmt MNs, we focused on structural changes in the MNs during in vitro maturation and aging. MNs were observed using microfluidic chambers (MFCs) (Fig. 2a, b). There was no obvious structural phenotype after 21 days of maturation (Fig. 2c–i), but this changed during longer in vitro aging (Fig. 2c–i). First, we identified significant increased axonal swelling followed by complete loss of motor axons at the distal exit site in FUSmt only (Fig. 2c, d). Following complete degeneration of distal axons, there was still no corresponding neuron loss at the proximal MFC site until 60 DIV (Fig. 2e–g); however, there were significantly increased caspase3-positive MNs (Fig. 2j–l, Supplementary Fig. S10). This was caused by the underlying FUS mutation as isogenic lines generated using the CRISPR/Cas9n technique showed significantly higher numbers of caspase3-positive MNs in FUSmt during cellular aging (Fig. 2l,

Supplementary Fig. S10). Cultivation for additional 50 days yielded an increased MN loss in FUS mutants (Fig. 2g–i).

Consistently, human postmortem tissue from FUS-ALS patients exhibited severe atrophy of skeletal muscles and

replacement of skeletal muscle parenchyma by connective and fat tissue (Fig. 3c) indicative of almost complete loss of skeletal muscle innervation. There was also considerable but not complete loss of lumbar spinal cord α-MNs with the presence of FUS

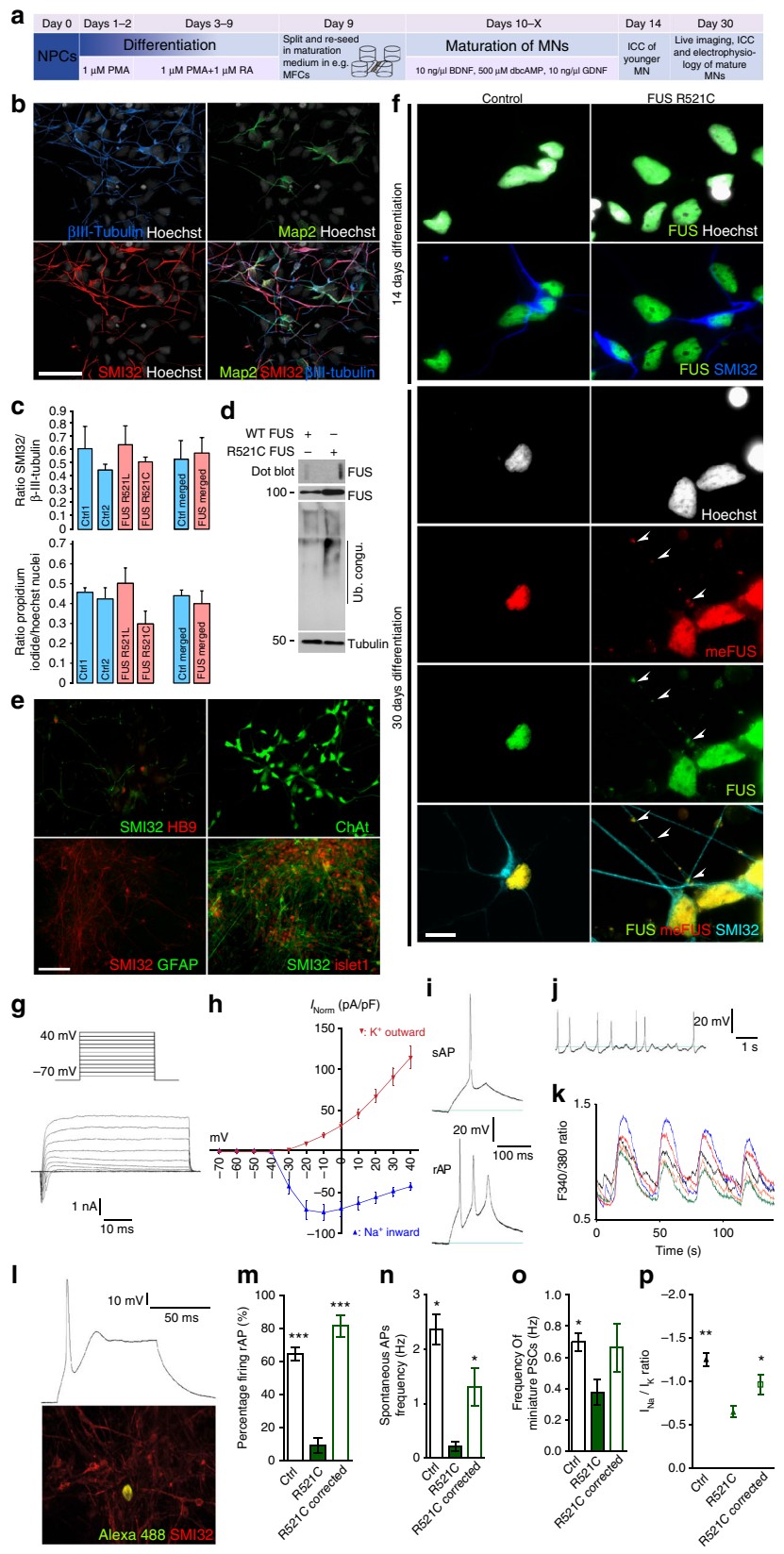

aggregates in surviving α-MNs (Fig. 3a, b-, e, f), and corresponding partial degeneration of ventral root axons (Fig. 3d). These data suggest that the neurodegenerative processes predominantly, however not exclusively, affected the distal axons, consistent with the results obtained with our in vitro model (Fig. 2). Thus, the data presented here with our iPSC-derived MN model are in agreement with a prominent distal axonopathy combined with a less severe neuronopathy in FUS ALS patients.

We next looked for early events in this neurodegenerative cascade. We performed live cell imaging of lysosomes and mitochondria between days 9–30 of maturation (Fig. 4), prior to the appearance of a structural phenotype (Fig. 2). We detected an overall reduction in organelle number (data not shown). In compartmentalized MN cultures using microfluidic chambers (MFCs), we observed the appearance of an axonal phenotype in FUS-ALS with a virtual arrest of mitochondria and lysosomes distally as opposed to normal motility proximally (Fig. 4a–i, Supplementary Movies 1–3, for detailed statistics of all box plots refer to Supplementary Tables 1–17). These axon trafficking defects were not detected in very early time points (9 DIV) but became obvious from 21 DIV onwards (Fig. 4a–c, Supplementary Movies 1 and 2). Analysis of Mitotracker JC-1—revealing the mitochondrial membrane potential—showed a loss of membrane potential only in the distal axon (Fig. 4d, g, Supplementary Movie 3) along with reduced mitochondria length (Fig. 4h), consistent with a recent report on non-neuronal cell models[29]. Lysosomes, however, remained normal in size (Fig. 4i). We confirmed that all control and FUS lines were phenotypically similar, thereby excluding clonal variability (Supplementary Figs. 2–9). The phenotypes were caused by the underlying FUS

### Table 1 Patient/proband characteristics

| | | Sex | Age at biopsy (years) | Mutation | Age at disease onset | Clinical phenotype | Disease duration (months) |
|---|---|---|---|---|---|---|---|
| Controls | hiPSC | Female | 48 | – | – | – | – |
| | | Male | 60 | – | – | – | – |
| | | Female | 45 | – | – | – | – |
| | | Female | 50 | – | – | – | – |
| | Autopsies | Female | 71 | – | – | – | – |
| | | Male | 70 | – | – | – | – |
| | | Male | 81 | – | – | – | – |
| | | Male | 80 | – | – | – | – |
| | | Male | 54 | – | – | – | – |
| | | Male | 67 | – | – | – | – |
| FUS-ALS | hiPSC | Female | 58 | R521C | 57 | Spinal | 7 |
| | | Isogenic control | | WT-GFP | | | |
| | | Isogenic mutant | | P525L-GFP | | | |
| | | Female | 65 | R521L | 61 | Spinal | 60 |
| | | Male | 29 | R495QfsX527 | n.d. | Spinal | n.d. |
| | Autopsies | Male | 40 | R521C | n.d. | Spinal | n.d. |
| | | Female | 70 | R521C | n.d. | Spinal | n.d. |
| | | Female | 35 | Y526C[49] | n.d. | Spinal/bulbar | n.d. |

n.d.: no data

**Fig. 1** Basal characterization of iPSC-derived FUS spinal MNs from ALS patients and controls. **a** Differentiation scheme of FUS and control MNs. **b** ICC of maturation markers (Map2, βIII-Tubulin, SMI32) highlighting MNs, bar: 50 μm. **c** Top: quantification of (**b**), bottom: quantification of cell death in mutant FUS versus WT controls, counts of pyknotic nuclei (ratio Propidium Iodide/Hoechst), $N = 3$, error bars = STDEV. **d** Western blots and dot blot analysis of WT (left lane) and FUS mutant (right lane; R521C), iPSC-derived MNs. Cell lysates were either subjected to dot blot analysis for FUS aggregates (top), or immunoblot analysis and probed with antibodies against FUS, ubiquitin, and tubulin. Note the augmented ubiquitination and aggregation of FUS in FUS MNs over WT. **e** Same as **b** but for more markers to further confirm MN identity: SMI32 (Neurofilament H), Hb9 (Homeobox gene 9), Cholinacetyltransferase (ChAt), Islet1 (ISL LIM homeobox 1). No glial cells were present in this cell culture indicated by the lack of GFAP- (glial fibrillary acid protein) positive cells, bar: 100 μm. **f** Aged FUS MNs displayed cytoplasmic, methylated FUS aggregates (white arrowheads), bar: 10 μm. **g–p** FUS mutant iPSC-derived MNs were functional and spontaneously active. **g** Illustration of stepwise depolarization in increments of 10 mV from a holding potential of −70 to 40 mV. **h** The potassium outward currents and sodium inward currents were normalized for the cell capacitance (mutant FUS cell line R521L, $n = 21$). **i** Recorded in the current-clamp mode during week 7 of differentiation, a majority of mutant FUS iPSC-derived MNs fired a single action potential (sAP; $90.7 \pm 9.3\%$; 19 out of 21 cells) upon stepwise depolarization or repetitive action potentials (rAP; $32.7 \pm 4.3\%$; 7 out of 21 cells). **j** Mutant FUS iPSC-derived MNs ($48.5 \pm 1.5\%$) were also spontaneously firing APs in varying frequencies ($0.77 \pm 0.19$ Hz). **k** Calcium-imaging analysis revealed the periodical spontaneous rise of intracellular calcium. **l** Patch clamp recordings of MN's. After recording, cells were filled with alexa 488, then labeled with neurofilament marker SMI32 validating MN identity. **m–p** Whole-cell patch-clamp data of MN recordings from healthy control (left bars, $N = 213$), mutant FUS (R521C.2, green middle bars, $N = 24$) and corresponding genetically corrected isogenic control line (R521C.2 corrected, right bars, $N = 16$) confirm the hypoexcitability phenotype of mutant FUS and suggest a functional recovery of the isogenic controls to healthy control level in terms of repetitive (**m**) and spontaneous APs (**n**), the frequency of post synaptic currents (PSCs) (**o**) as well as the Na$^+$/K$^+$-ratio (**p**). Data are plotted as mean, statistical comparison of healthy and isogenic control with mutant FUS using an unpaired t-test, *, **, ***P values of 0.05, 0.01, and 0.001, respectively, error bars = STDEV (**c**) or SEM (**h**, **m–p**)

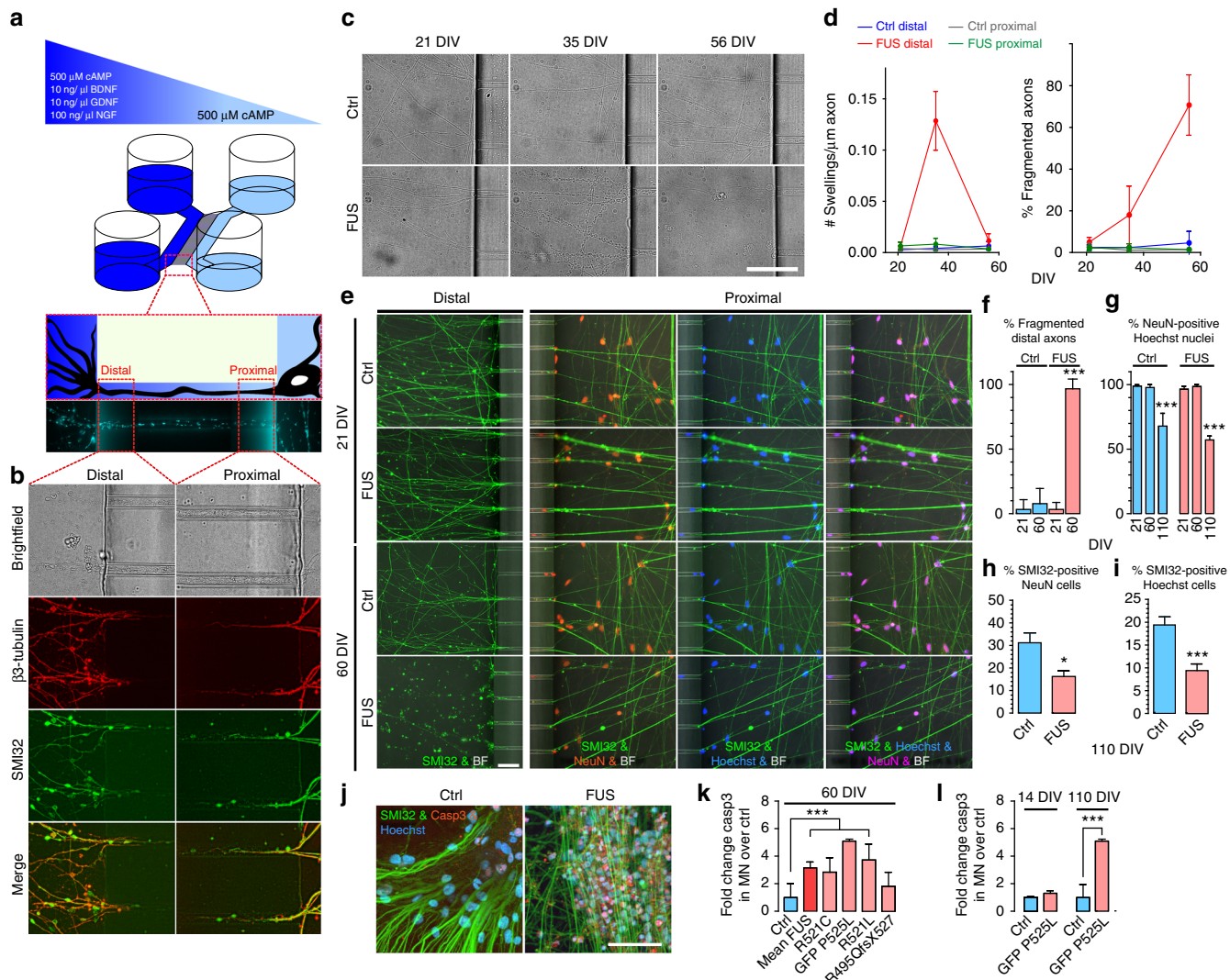

**Fig. 2** Sequential development of neurodegeneration in FUS-ALS MNs. **a** Cells were differentiated and afterwards seeded to one side (light blue). Directed growth of axons through 900 μm long microchannels to the other side (dark blue) was realized by the application of a volume and maturation factor gradient functioning as a guidance cue. For a standardized statistical readout the "Distal" and "Proximal" Windows were defined. **b** Phase contrast and ICC of MN (representative example: Ctrl1, Table 1) cultivated in the MFCs. All grown-through axons were positive for βIII Tubulin and SMI32, labeling them as originated from MNs, bar: 20 μm. **c** Brightfield (BF) images acquired over aging of MN at the distal MFC exit revealed prominent axonal swelling and fragmentation beyond 21 DIV in FUS-GFP P525L as opposed to FUS-GFP WT control cells, bar: 50 μm. **d** Quantification of **c**, left: count swellings, right: % fragmented distal axons, N = 10, error bars = STDEV. **e** IF images acquired over aging of MN at the distal exit versus proximal entry site with distal stainings for SMI32 (MN marker Neurofilament H) and proximal stainings for NeuN (neuronal nuclear marker) and Hoechst, Ctrl: FUS-GFP WT, FUS: FUS-GFP P525L. Note the complete loss of SMI32-positive distal axons (i.e. fragmentation) in FUS at 60 DIV as opposed to healthy control cells whereas the soma and network integrity remained intact at the proximal site, indicating a dying-back in FUSmts, bar: 50 μm. **f–i** Quantification of **e** and in addition at 110 DIV, **f** distal axon integrity (% fragmented SMI32-positive axons), **g** % NeuN-positive Hoechst nuclei, **h** % SMI32-positive NeuN cells and **i** % SMI32-positive Hoechst cells, N = 10. **j** Increased apoptosis in FUS-GFP P525L MNs at 60 DIV as indicated by Caspase 3 IF staining (see Supplementary Fig. 10 for detailed gallery). **k** Quantification of **j** and all other mutant FUS MNs (see Table 1) at 60 DIV, N (ctrl) = 31, N (mean FUS) = 41, N (R521C) = 8, N (GFP-P525L) = 2, N (R521L) = 10, N (R495QfsX527) = 10, error bars = STDEV. **l** Isogenic ctrl/P525L MNs show increase of Caspase 3 MNs in the mutant line during cellular aging (14 versus 110 DIV), N = 8. All data are plotted as mean, statistics (**f–i**, **k**, **l**): one-way ANOVA with post-hoc Bonferroni test (*, **, ***P values of 0.05, 0.01, and 0.001, error bars = STDEV)

mutation as isogenic controls generated using the CRISPR/Cas9n technique presented a normal phenotype (Supplementary Fig. 1, Fig. 4e, Supplementary Movie 3).

**DNA damage causes neurodegeneration in FUS-ALS iPSC MNs.** Recent evidence suggests that DNA damage occurs in animal models of FUS[19] and in patients with FUS-ALS[14]. Therefore, we tested for the occurence of DNA double strand breaks (DSBs) in our hiPSC-derived FUS-ALS MN model. Immunofluorescence analysis suggested significantly increased DSBs (γH2AX immunoreactivity) in FUS-ALS-derived neuronal cells (Fig. 5a–d), in fact increased DSBs were evident in both mature spinal MNs and in the neural progenitor cells (Fig. 5a–d), suggesting that these observations correspond to an early event in the FUS-ALS pathophysiology. This is strengthened by the fact that increased DSBs were already visible in MNs on DIV14 (Fig. 5b) prior to cytoplasmic FUS mislocalization (Fig. 1f). There

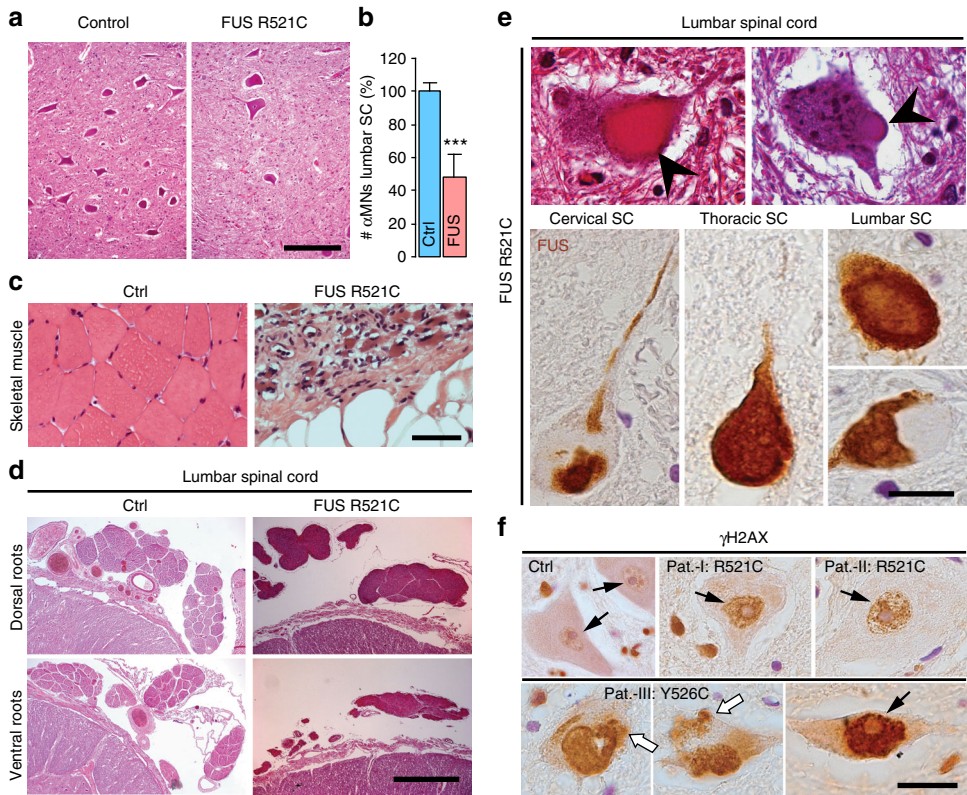

**Fig. 3** Human FUS lumbar spinal cord and skeletal muscle histology. **a** Human FUS lumbar spinal cord histology: reduced α-MNs in the ventral horn, bar: 200 μm. **b** Quantification of **a** from controls ($n = 5$) compared to the FUSmts ($n = 4$) cases. Quantification done on H&E staining; 5 consecutive paraffin sections per case. Data are plotted as mean, error bars = STDEV, unpaired $t$-test (*, **, ***$P$ values of 0.05, 0.01, and 0.001, respectively). **c** Severe atrophy of skeletal muscles and replacement of skeletal muscle parenchyma by connective and fat tissue in FUSmts, bar: 200 μm. **d** Ventral versus dorsal roots atrophy, bar: 900 μm. **e** FUS aggregates in lumbar spinal cord α-MNs (arrow in upper panel), HE staining, bar: 40 μm; lower panel, FUS-inclusions in α-MNs (different spinal cord levels: cervical, thoracic, lumbar), bar: 30 μm. **f** Representative IHC of human ALS-FUS cases with DSB marker γH2AX antibody (black arrows), of lumbar spinal cord α-MNs. Note the peculiar cytoplasmic staining of γH2AX in one particular case (white arrows). Paraffin section, bar: 50 μm

were significantly more DSBs in the P525L line compared to the isogenic control line (Fig. 5e). Similar to untreated conditions, DSBs were significantly increased in P525L neurons compared to isogenic controls 24 h after treating with etoposide for 1 h (Fig. 5e). Interestingly, P525L neurons were still able to repair DNA damage as shown by recovery experiments after etoposide treatment (Fig. 5e, f).

α-MNs of lumbar spinal cord from FUS-ALS patients (Table 1) consistently showed DNA DSBs as suggested by a robust immunoreactivity of γH2AX in the nucleus (Figs. 3f and 5n, black arrows). Surprisingly, however, a few α-MNs from one patient (Y526C) showed increased γH2AX labeling both in the nucleus (Fig. 3f; black arrows) as well as in the cytoplasm (Fig. 3f; white arrows). Immunofluorescence analysis confirmed the aberrant co-localization of γH2AX accumulations with FUS aggregates in the cytoplasm of surviving α-MNs (Fig. 5o, yellow arrow) in the lumbar spinal cord of this particular patient. Taken together, our results support the notion that DNA damage is probably an early event in FUS-ALS.

We next asked whether DNA damage is the cause or consequence of FUS-ALS pathophysiology[30]. First, we induced DNA damage in spinal MNs derived from healthy controls. Proximal (soma site) etoposide treatment increased DNA DSBs in controls dramatically (Fig. 5b, d). In addition, this caused a loss of mitochondrial and lysosomal motility in distal axons (Fig. 5h–j, Supplementary Movies 4 and 5), a drop in mitochondrial membrane potential (Fig. 5h, k, Supplementary Movie 4) and

mitochondrial fragmentation (Fig. 5l). These effects were absent in the proximal axon parts (Supplementary Figs. 3–5, Supplementary Movies 4 and 5), thus mimicking the FUS-ALS phenotype (Fig. 4). Importantly these effects were also absent when etoposide or arsenite was added to distal axons only (Supplementary Figs. 3–5, Supplementary Movies 4 and 5). Finally, etoposide treatment of hiPSC-derived MNs leads to FUS mislocalization with the appearance of cytosolic FUS inclusions (Figs. 5b and 6e), confirming that the DNA damage is upstream of neurodegeneration and aggregate formation.

**Impaired FUS shuttling is the upstream event in FUS-ALS.** To further substantiate the link between DDR, FUS aggregation and neurodegeneration, we investigated various aspects of DNA repair signaling in hiPSC-derived MNs (Figs. 6 and 7). At first, we asked whether NLS mutations in the *FUS* gene are sufficient to impair recruitment to DDS in patient-derived human MNs, as recently reported in heterologous (over-)expression models[12–14,19]. For this, we generated isogenic iPSCs with a carboxyterminal GFP tag on the endogenous FUS protein (wildtype (wt) and mutant (mt) P525L) (Supplementary Fig. 1, Table 1). Laser microirradiation caused fast and transient recruitment of FUS-GFP to DNA damage sites in wt MNs (Fig. 6a, b, Supplementary Movie 6), however, this process of FUS recruitment was diminished in FUS-NLS mutant lines (Fig. 6a, b, Supplementary Movie 6) consistent with the recently reported studies using heterologous expression in

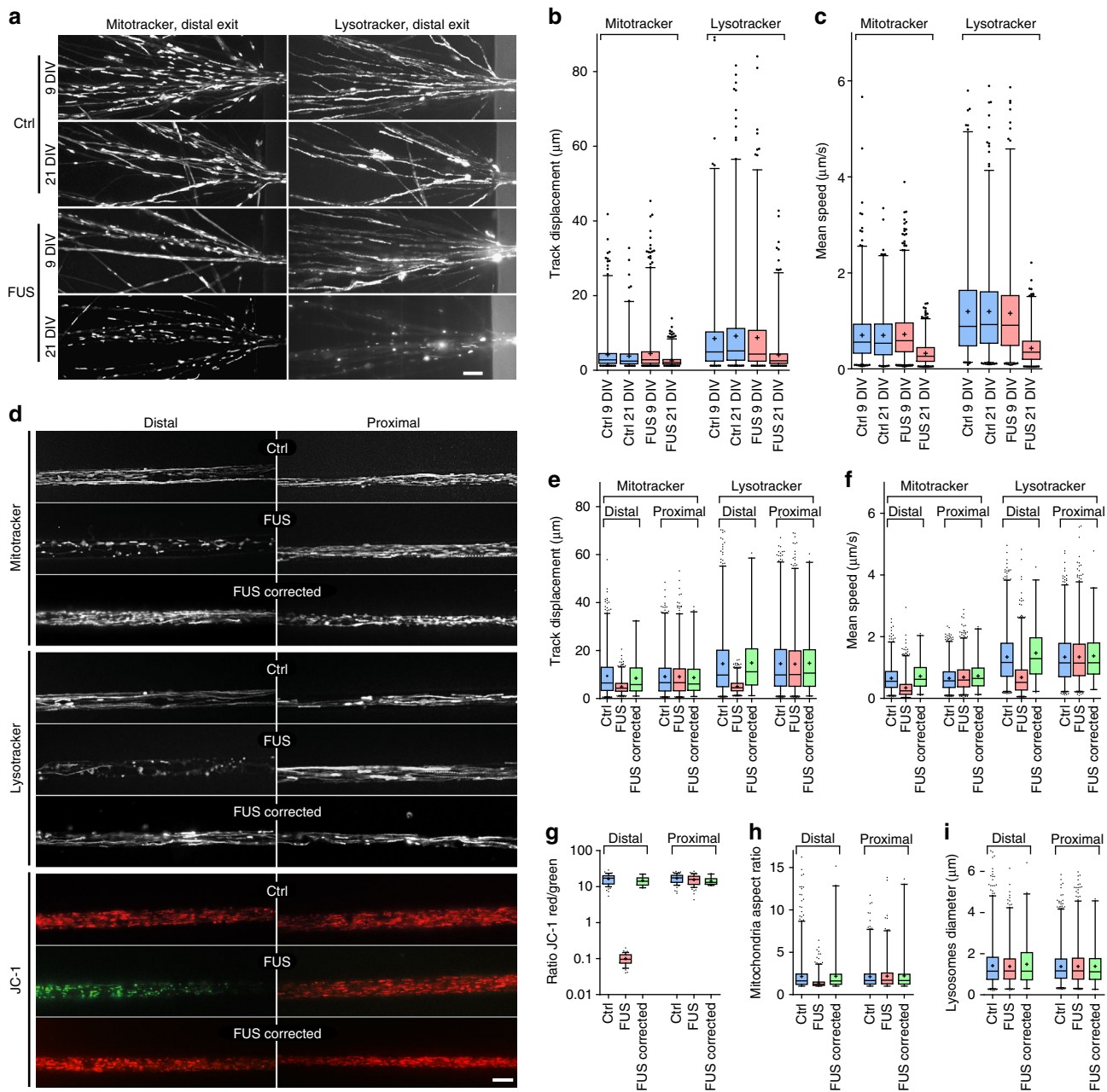

**Fig. 4** MNs develop trafficking defects of mitochondria and lysosomes over time along with loss of mitochondria function in distal axons. **a** FUS MN movies of shorter 150 μm-MFC's at earlier maturation stage (9 DIV) versus standard endpoint time (21 DIV). Shown is the motility of mitochondria (left, Supplementary Movie 1) and lysosomes (right, Supplementary Movie 2) at the distal exit as maximum intensity projection. **b**, **c** Organelle tracking analysis as box plots, Ctrl: FUS-GFP WT, FUS: FUS-GFP P525L as described in Table 1. **d** FUS MN movies: distal organelle arrest and loss of mitochondrial membrane potential at 21 DIV in 900 μm-MFC's (representative examples: Ctrl1 versus FUS R521C), bar: 10 μm (Supplementary Movie 3). Note the rescue in isogenic FUS R521C corrected control cells. Shown is maximum intensity projection except for JC-1 movies, in which first frame of movie is shown. **e–i** Organelle tracking (**e**, **f**), mitochondrial membrane potential (**g**) and shape (**h**, **i**) analyses as box plots, batch analysis of Ctrl1-3 (Ctrl) and FUS R521C, R521L, R495QfsX527 (FUS) as described in Table 1

HeLa cells[12,13,17], U2OS cells and murine primary neurons[14]. In fact, this inhibition was already observed in neural progenitor cells (NPCs) (Supplementary Movie 7), implying that this is an early upstream event. Thus, FUS-NLS mutations also impair FUS recruitment to DDS in patient-derived MNs.

We next asked whether interfering with nucleo-cytoplasmic shuttling of FUS is sufficient to impair DNA damage sensing by FUS. Earlier work has shown that FUS mislocalization depends on nucleo-cytoplasmic shuttling[4] which in turn depends on two main pathways: Transportin (TRN)-mediated nuclear import of

FUS, known to be disrupted in FUS-NLS mutations[4–6], and DNA-PK mediated nuclear export secondary to DNA damage induction[8]. Arginine methylation of the PY-NLS domain modulates TRN binding to FUS and its nuclear import and inhibition of arginine methylation is known to restore TRN-mediated nuclear import in FUS-NLS mutant HeLa models[5]. FUS+ aggregates in ALS postmortem specimens were reported to contain methylated FUS[5], consistent with our iPSC-derived MNs (Fig. 1f). To test whether restoration of nuclear import of FUS can rescue its impaired recruitment to DDS, we

performed chemical inhibition of arginine methylation using adenosine-2,3-dialdehyde (AdOx)[5]. Indeed, AdOx rescued FUS mislocalization (Fig. 6e, f) and recruitment to DDS (Fig. 6c, d, Supplementary Movie 6), but also distal axon trafficking (Fig. 6g–l,

Supplementary Movies 8 and 9). Nuclear export of FUS depends on DNA-damage-induced FUS phosphorylation by DNA-PK[8]. Consistently, inhibition of DNA-PK (NU7441) also restored FUS cytosolic mislocalization (Fig. 6e, f), recruitment to DDS (Fig. 6c,

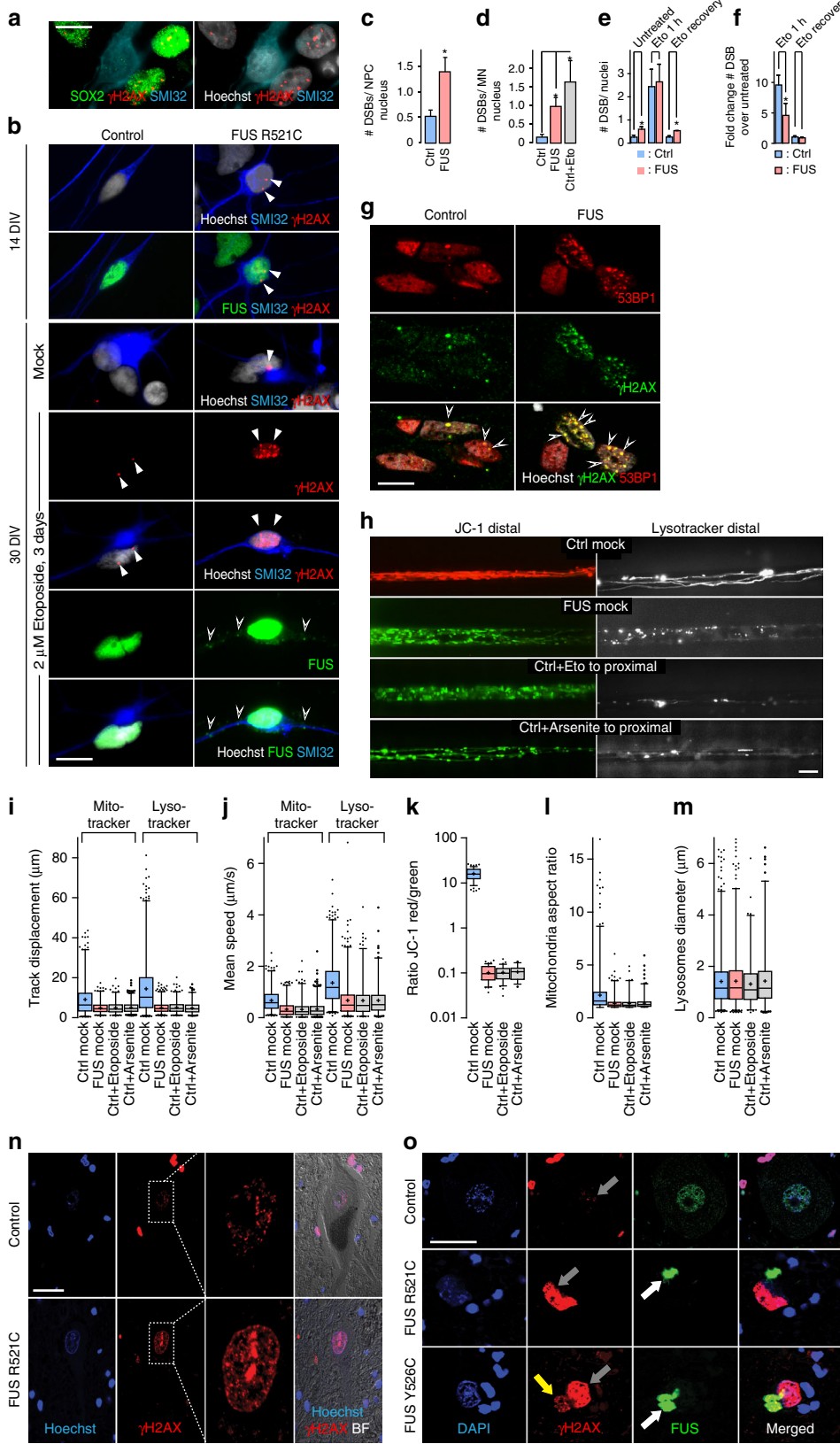

d, Supplementary Movie 6) and distal axon trafficking (Fig. 6g–l, Supplementary Movies 8 and 9). Taken together, these results provide strong evidence that impairment of nucleo-cytoplasmic shuttling is responsible for aggregate formation, the impairment of FUS-dependent DDR and neurodegeneration.

**FUS NLS pathology relies on PARP-dependent FUS-mediated DDR.** We next asked whether induction of DNA damage is sufficient to induce cytoplasmic mislocalization of FUS. Chemically inducing or increasing DNA damage by either by etoposide or arsenite (the latter known as a ROS-mediated DNA damage inducer) in untagged (Fig. 5b, d) or GFP-tagged human MNs (Fig. 6e, f) actually augmented cytoplasmic FUS mislocalization consistent with our above observations. These findings also correspond with recent data on murine cells showing an accumulation of FUS in the cytoplasm after DNA damage induction due to activation of DNA-PK[8]. Consistently, we also observed inhibition of FUS recruitment to DDS due to etoposide/arsenite treatment of control MNs (Fig. 6c, d, Supplementary Movie 6). PARylation is a crucial event in DNA repair. DNA damage is the primary activator of PARP1 that catalyzes the reaction of poly (ADP-ribosylation) (for review see ref. [15]). We thus investigated if the observed FUS NLS phenotypes rely on the PARP-dependent DDR signaling[12–14,17]. Indeed, PARP1 inhibition caused a reduction of FUS recruitment to DDS in control cells (Fig. 6c, d, Supplementary Movie 6) as previously reported for HeLa cells[17]. Furthermore, PARP1 inhibition also led to cytoplasmic FUS aggregation (Fig. 6e, f) and defects in distal axonal trafficking (Fig. 5g–l, Supplementary Movies 8 and 9), thereby faithfully mimicking FUS-NLS-mutant phenotypes.

PAR is degraded by poly(ADP-ribose) glycohydrolase (PARG)[18] and PARG inhibition leads to prolonged persistence of FUS at DDS[17]. In our experimental setup, PARG inhibition restored FUS recruitment to DDS in FUSmt MN (Fig. 6c, d, Supplementary Movie 6). Moreover, PARG inhibition was sufficient to prevent FUS aggregate formation (Fig. 6e, f) and completely restored the defective distal axon trafficking phenotype (Fig. 6g–l, Supplementary Movies 8 and 9). It is unlikely that NLS mutations impair binding to PAR since the RGG domain was recently identified as being responsible for PAR binding to FUS[13,25]. We confirmed these findings and showed that prolonged PAR activity by PARG inhibition restored the NLS mutant phenotypes (Fig. 6g–l). These results are consistent with the idea that the FUS NLS pathophysiology actually relies on the PAR-dependent FUS-mediated DDR signaling subsequently leading to cytoplasmic FUS aggregation and neurodegeneration.

Only treatment (PARP1-, PARG inhibitor, etoposide, arsenite) at the proximal soma site, and not at the distal axon, induced or reversed the defective distal trafficking phenotype (Supplementary Figs. 3–9, Supplementary Movies 4, 5, 8 and 9). Compounds added exclusively to the proximal site cannot migrate to the distal site because the microflow in the microfluidic chambers is directed from distal to proximal (due to the higher distal liquid level, Fig. 2a and Methods). Therefore, the impact of the proximal treatments on distal axon trafficking occurred in the physical absence of these compounds, thereby excluding the possibilities of a local mode of action.

To further strengthen these observations, we tried to induce local (distal) axonal trafficking inhibition by either nocodazole (known to depolymerize microtubules) or oligomycin A treatment (complex 5 inhibitor). Both treatments caused local axonal trafficking arrest and axonal swelling (Fig. 6m, Supplementary Movies 10 and 11), but neither of them leads to deficits in FUS recruitment to DSBs nor FUS cytoplasmic mislocalization and aggregation formation (Fig. 6n, o). In conclusion, the mimic/rescue of distal axonal phenotypes must be due to remote, upstream manipulation of DNA repair mechanisms, arguing against a pure axonopathy and in favor of a neuronopathy with prominent and early axonal degeneration (see also Figs. 2c–i and 4a–c) as starting point of the neurodegenerative disease process.

**DNA damage enhances FUS mislocalization and neurodegeneration.** Given that DNA damage recruits FUS and that FUS mislocalization disrupts DDR signaling, we hypothesized that FUS-NLS mutations induce a vicious cycle whereby DNA damage subsequently induces additional cytoplasmic FUS mislocalization which in turn results in aggravated DDR signaling through a consecutive loss of nuclear function. Time lapse experiments showed time-dependent cytosolic FUS mislocalization and appearance of cytoplasmic FUS inclusions after interference with DDR signaling (PARP1 inhib.) or DNA damage induction (etoposide) (Fig. 7a). While recruitment to DDS was blocked before the appearance of cytosolic FUS mislocalization by PARP1 inhibition in FUSwt MNs (Fig. 7a–c, Supplementary Movie 12), this was seen by etoposide treatment only after the appearance of cytosolic FUS mislocalization (Fig. 7a–c, Supplementary Movie 12). To further test the vicious cycle hypothesis, we performed several co-treatments. While DNA-PK inhibition or AdOX was able to restore the lack of FUS recruitment to DDS and FUS mislocalization by etoposide or arsenite (Fig. 7d, e, Supplementary Movie 13), DNA-PK inhibition did not rescue FUS recruitment to DDS during PARP1 inhibition (Fig. 7d, e) but did completely restore cytoplasmic mislocalization (Fig. 7f). This

**Fig. 5** DDR signaling is involved in FUS-NLS pathophysiology. **a** IF of SOX2 staining (green) validated NPC identity for DSB counting (by γH2AX, red) in (**c**). **b** DNA DSBs in iPSC-derived FUS R521C spinal MNs (right) versus WT control cells (Ctrl1 left, Table 1) by ICC for nuclear γH2AX foci (red) at 14 and 30 DIV, bar: 10 µm. More DSB foci (white arrowheads) were observed in FUS mutants (right) already at 14 DIV and increased in controls (left) and FUS mutants after Etoposide treatment. In addition, cytoplasmic FUS inclusions in FUS mutants increased after etoposide treatment (green, hollow arrowheads). **c, d** Both, FUS-NPCs (**c**) and mature–MNs (**d**) showed more DSBs per nucleus over controls (N = 3). **e** Untreated mature FUS-GFP P525L (FUS) MN showed increased nuclear DSB foci over isogenic control cells, consistent with untagged FUS lines in (**d**). 1 h etoposide treatment and 24 h withdrawal led to a similar response compared to control cells (i.e. transient increase of DSB foci and reversion to basal levels indicative of successful DSB repair). **f** Same as **e** but expressed as fold change over respective untreated control (N = 4). Statistics **c–f**: data are plotted as mean, unpaired t-test (only **c**) or one-way ANOVA (**d–f**) with post-hoc Bonferroni test (*, **, ***P values of 0.05, 0.01, and 0.001, respectively, error bars = STDEV). **g** Validation of the anti-γH2AX antibody in FUS-GFP P525L MNs (without etoposide) by costaining with anti-53BP1, a second marker for DSBs. Note the prominent colocalization (merge, yellow overlapping). **h** Live imaging of MN at 21 DIV shown as maximum intensity projections: distal loss of organelle motility and mitochondrial membrane potential in FUS mimicked through proximal etoposide or arsenite addition to control, bar: 10 µm (Supplementary Movies 4, 5). **i–m** Organelle tracking (**i, j**), mitochondrial membrane potential (**k**) and shape (**l, m**) analyses of (**h**) as box plots, batch analysis of Ctrl1-3 (Ctrl) and FUS R521C, R521L, R495QfsX527 (FUS) as described in Table 1, exception: arsenite treatment only on FUS corrected R521C (FUS-GFP WT). **n** ICC showing augmented DSBs (γH2AX) in ALS-FUS case over healthy control person, bar: 20 µm. **o** ICC showing FUS aggregation in the cytoplasm (white arrows) of a typical ALS-FUS case with R521C mutation (mid gallery row) in the NLS along with augmented γH2AX occurrence in the nucleus (gray arrows) as opposed to an atypical, peculiar case (Y526C, bottom gallery row) with cytoplasmic co-aggregation of FUS and γH2AX (yellow arrow), bar: 20 µm

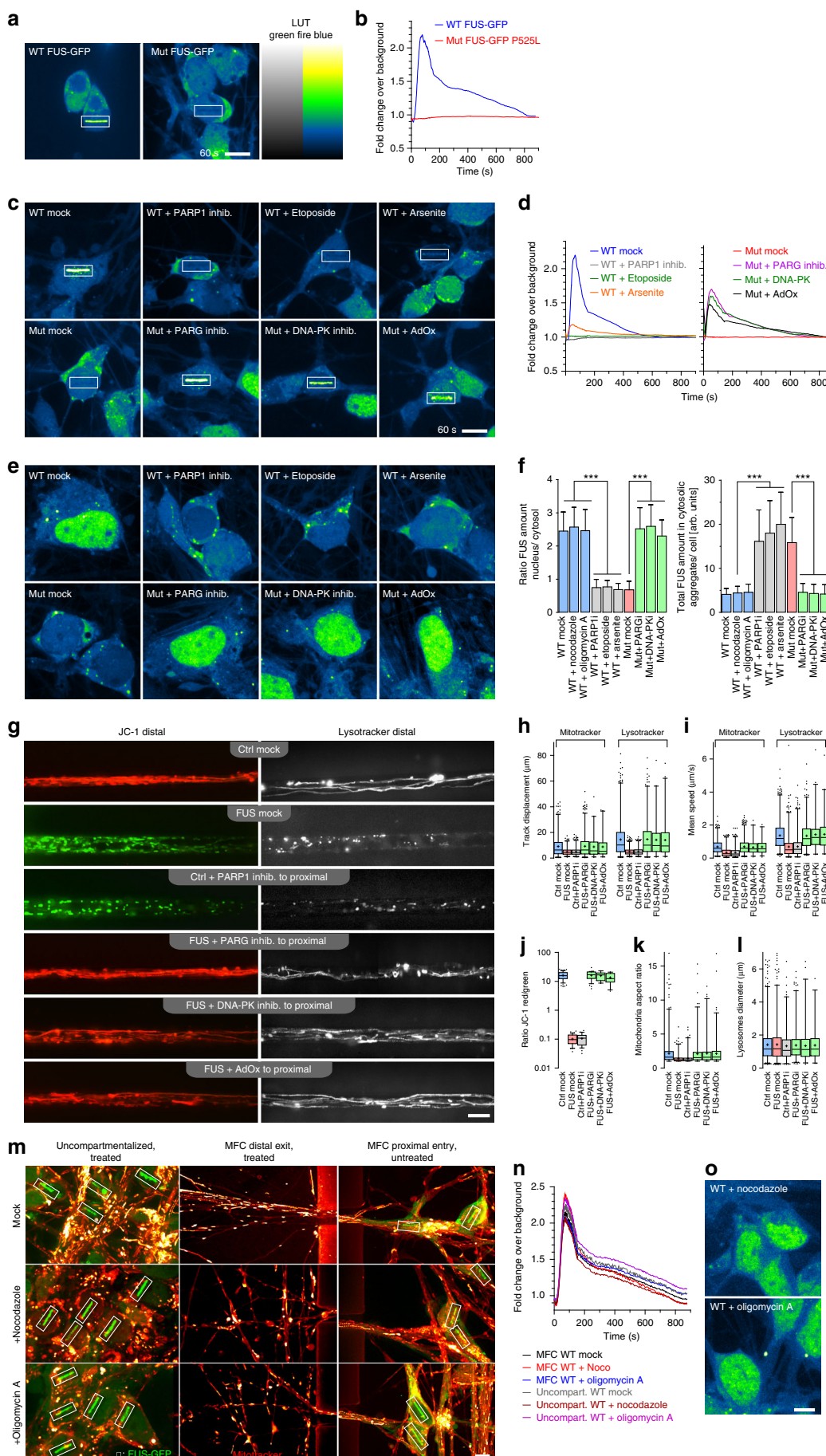

shows that DNA-PK is downstream of DNA damage response signaling induction. It further suggests that DNA-PK's strongest effect in FUS-ALS is the regulation of FUS nucleo-cytoplasmic shuttling[8].

To gain further insights into possible downstream mechanisms in DDR by FUS we analyzed the two major pathways of DDR in postmitotic cells, namely classical non-homologous end joining (c-NHEJ) and alternative non-homologous end joining (a-NHEJ). While c-NHEJ depends on DNA-PK activation, a-NHEJ pathway is downstream of PARP1[31]. Within the c-NHEJ pathway, we found significant higher levels of total DNA-PK consistent with increased basal DDS in FUSmt MNs with no changes in phospho/total-DNA-PK, and no differences in KU70/KU80 (Fig. 7g, h, Supplementary Fig. 12). There was also no significant difference in the levels of a-NHEJ proteins (LIG1, PARP1) (Fig. 7g, h, Supplementary Fig. 12). DNA damage induction (etoposide) in FUSwt and FUSmt MNs resulted in a similar increase of phospho-/total DNA-PK and cleaved /full-length PARP1 ratio (Fig. 7g, i, Supplementary Fig. 12) suggesting no major disturbance of the downstream DNA damage response machinery in FUSmts (Fig. 5d, e). In turn, DNA-PK inhibition led to decreased phospho/total DNA-PK ratio but increased full length PARP1 (Fig. 7g, i, Supplementary Fig. 12), consistent with more active PARP1 in FUSmt MNs as if treated with PARG inhibition. These data underpin that DNA-PK serves mainly in FUS nucleo-cytoplasmic shuttling and that DDR mainly relies on poly(ADP-ribose) dependent a-NHEJ. In summary, we show that DNA damage is a key event in FUS-ALS which leads to DNA-PK activation. In turn, DNA-PK activation enhances cytoplasmic localization of FUS, thereby closing the vicious cycle by additionally depleting nuclear FUS, which is already the case in ALS-causing FUS-NLS mutations by impaired TRN-mediated nuclear import.

## Discussion

Using human iPSC-derived MNs and human postmortem specimens we show that FUS-ALS is caused by impairment of proper DNA damage response signaling subsequently leading to neurodegeneration and aggregate formation. Currently, there is much attention given to mechanisms of protein aggregation and to their clearance[17,23,24]. In contrast, our work suggests the need of novel therapeutic pathways upstream of aggregate formation involving modulation of DNA damage pathways. Some of these compounds

modulating DDR are already in clinical trials for cancer therapy and could thus rapidly be adapted to ALS.

There have been hints that DNA damage is apparent in FUS-ALS and that FUS-NLS mutations impair recruitment of FUS to DNA damage sites[12], while others report only marginal[13] or no obvious phenotype[14] in DNA damage recruitment by NLS mutations. These differences most likely arise from the different cell types used in the respective studies (hiPSC-derived MNs were only used in our study) but might also be due to the technique of FUS expression (ectopic expression compared to endogenous expression, the latter used for the first time in the current study), although the expression levels were carefully controlled in the previous studies[14].

Furthermore, PARP is involved in forming liquid compartments of FUS at sites of DNA damage, and aberrant phase transition of the liquid compartments to solid-like aggregates could be involved in the onset of the disease[17,23,24]. However, the exact relationship between DNA damage and the formation of cytoplasmic aggregates as well as neurodegeneration was lacking. Here, we show that DNA damage enhances cytoplasmic FUS mislocalization, thereby inducing a vicious cycle, in which failure of DNA damage repair signaling further enhances FUS mislocalization and induces aggregation and neurodegeneration.

Our study thus adds FUS-ALS to the class of neurodegenerative diseases with impaired DDR signaling, such as Ataxia teleangiectatica, AOA1 and SCA3. Interestingly, FUS$^{-/-}$ mice suffer from genomic instability[22] and enhanced radiation sensitivity[32]. Whether this holds true for FUS-ALS patients is currently unknown. Furthermore, Parp1$^{-/-}$ mice were reported to suffer from high energy expenditure and decreased body fat mass similar to ALS patients (for review see ref. [33]). This might be clinically relevant in many respects. For example, PARP1 inhibition has recently been suggested as a therapeutic strategy for neurodegenerative diseases (for review see ref. [15]). In contrast, our data strongly argue against PARP1 inhibition in FUS-ALS, but suggest PARG or DNA-PK inhibition as promising treatment strategies.

Another link connecting FUS and DDR came from two reports on the interaction of FUS and HDAC1. Both reports showed that FUS directly interacted with HDAC1 and that this interaction is important for proper DDR. Consistently, FUS-NLS mutations showed a diminished interaction with HDAC1[14,19]. Furthermore, FUS-NLS mutant mice showed shortened dendrites at least in part due to BDNF signaling deficiency mediated by DNA damage

**Fig. 6** DDR signaling impairment in FUS-NLS mutations is upstream of neurodegeneration and aggregation formation. **a** Mutant P525L FUS failed to be recruited to DNA damage sites. Recruitment-withdrawal to Laser cuts in MN nuclei (boxed area) expressing normal (WT) or mutant P525L (Mut) FUS-GFP was imaged live at 21 DIV (Supplementary Movie 6). **b** Quantification of **a**, FUS-GFP at cut over time. **c** Chemical modulation of nuclear FUS impacted on recruitment of FUS-GFP to cuts in WT cells (top gallery) treated 24 h mock, with PARP1 inhib., etoposide or arsenite or in mutant cells (bottom gallery) treated 24 h mock, with PARG, DNA-PK inhibitors or AdOx (Supplementary Movie 6). **d** Quantification of **c**. **e** Compounds impacted on nuclear FUS-GFP levels in MNs: treatments WT versus Mut as for **c**, bars: 10 μm. **f** Quantification of **e**, ratio FUS-GFP amount nucleus/cytosol (left) or total FUS amount in cytoplasmic aggregates (right). Data are plotted as mean, error bars = STDEV, one-way ANOVA with post-hoc Bonferroni test (*, **, ***P values of 0.05, 0.01, and 0.001, respectively, N = 3). **g** Proximal compound addition impacted on distal axonal trafficking at 21 DIV depicted as maximum intensity projections: PAPR1 inhib. mimicked mtFUS-phenotype whereas PARG, DNA-PK inibitors and AdOx rescued mtFUS phenotype (Supplementary Movies 8, 9). Shown are representative examples of Ctrl1 and R521L, see Table 1. **h–l** Organelle tracking (**h**, **i**), mitochondrial membrane potential (**j**) and shape (**k**, **l**) analyses of G as box plots, batch analysis of Ctrl1-3 (Ctrl: Mock, PARP1 inhib.) and FUS R521C, R521L, R495QfsX527 (FUS: PARG inhib.) as described in Table 1, exception: DNA-PK and AdOx inhib. only on FUS R521L. **m** Left column: impact of microtubule disruption (24 h nocodazole, 5 μM, mid) or respiratory inhibition (24 h oligomycin A, 10 μM, bottom) on the recruitment of WT FUS-GFP to the Laser cut in nuclei (boxed areas) in uncompartmentalized MN at 21 DIV (Supplementary Movie 10). Note the unaltered FUS recruitment (green) despite the severe disruption of the mitochondria network (mitotracker deep red FM, LUT red hot) along with loss of processive motility in the treated cells (maximum intensity projection of movies). Mid and right column: ditto for treatment exclusively at the distal exit site of 900 μm-MFCs (Supplementary Movie 11). Note the unaltered FUS recruitment (proximal entry, boxed areas) despite the severe disruption of the mitochondria network along with loss of processive motility at the distal site, bar: 10 μm. **n** Quantification of FUS-GFP recruitment in **m**. **o** Nocodazole or oligomycin A treatment in **m** did not alter the normal nuclear FUS-GFP localization

induction[19]. FUS-NLS mutations most likely do not cause a complete loss of function of FUS protein since treatment with PARG is sufficient to restore FUS recruitment to DNA damage sites, cytoplasmic mislocalization and axon trafficking phenotypes (Fig. 6c, e, g). These findings are actually consistent with the study

by Wang et al. showing that mutant FUS proteins are still—at least in part—recruited to DDS, but were also impaired in the later steps of assembly or stabilization of the repair complex[14].

FUS was shown to be required for DSB repair by homologous recombination (HR) or non-homologous end joining (NHEJ) and

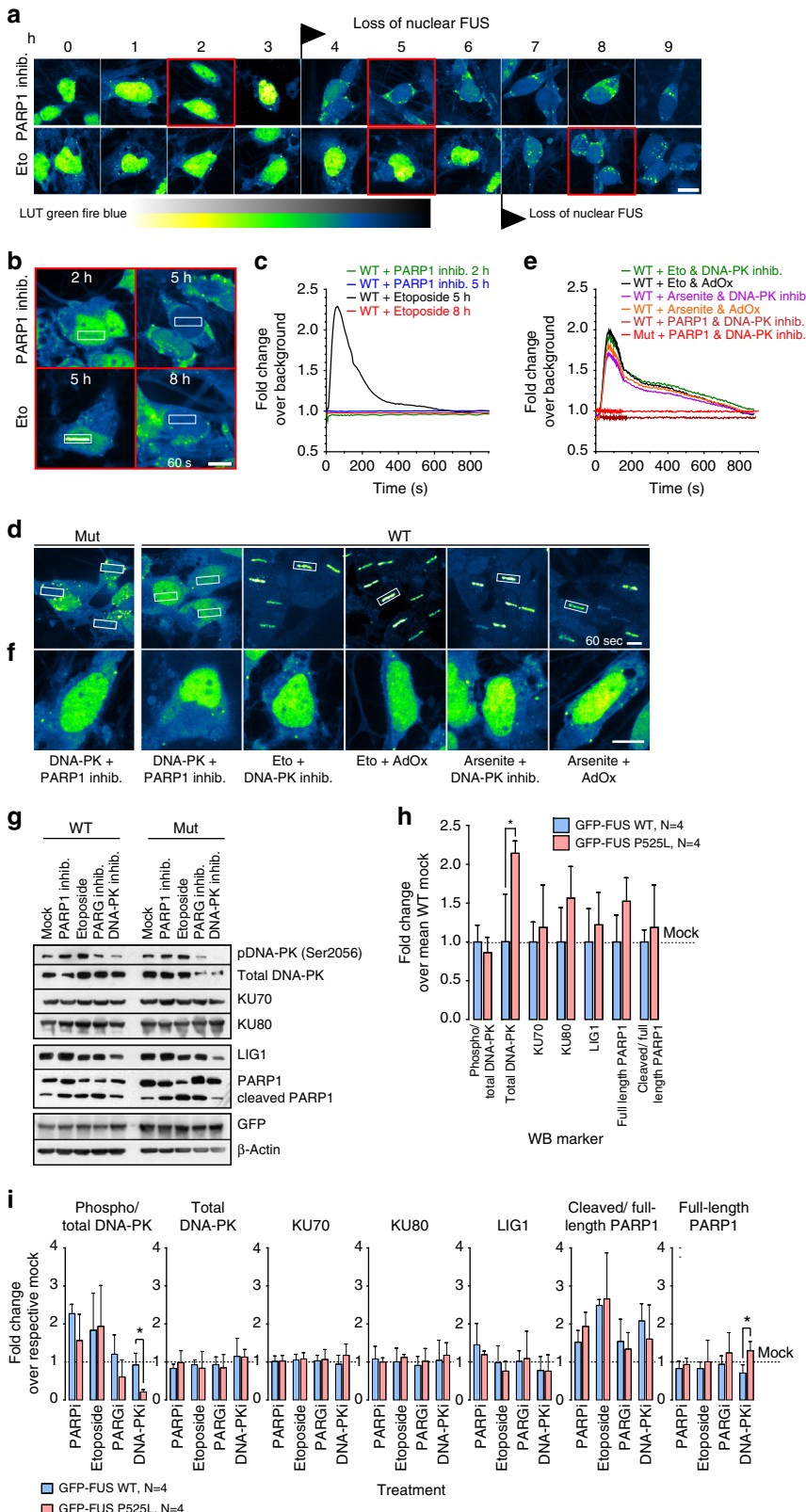

FUS knockdown reduced both HR and NHEJ efficacy[13,14]. Furthermore, NHEJ efficacy was reduced by overexpression of FUS R521C but not by wtFUS protein in murine primary cortical neurons[14]. Interestingly, FUS knockdown affected neither the c-NHEJ factors KU70 and DNA-PK nor ATM substrates, such as 53BP1[13], consistent with our data suggesting poly(ADP-ribose) mediated a-NHEJ as the DDR pathway downstream of FUS. Moreover, FUS was reported to directly interact with PAR[13] and PARylation was shown to induce additional PARP1 recruitment to DDS[20]. Thus, it might be possible that FUS mutations interfere with this interaction and might impair this feed-forward loop. However, the detailed downstream pathways for FUS mediated DDR signaling and its possible disturbance by FUS mutations needs further examination.

FUS was initially characterized as an oncogene (fused in sarcoma)[34]. FUS-CHOP fusion proteins lacks the c-terminal region, including RGG and NLS domain[13] and are responsible for PAR binding[13] and DNA DSB repair ([14,19] and our study). Interestingly enough, FUS-CHOP-positive myxoid liposarcoma are radio-sensitive[35] and FUS knockdown increases radiosensitivity[13]. Furthermore, Ews−/− mice, another protein of the FET family (of which FUS is a member), are also highly sensitive to irradiation[36]. This is of immense clinical impact; currently, radiation is used on salivary glands as a treatment of pseudohypersalivation in ALS patients.

For dynamic measurement of FUS mislocalization, DNA recruitment and protein aggregation we generated isogenic iPSC lines with GFP-tags of the endogenous FUS protein using CRISPR/Cas9n. Thus, the results obtained herein are clearly due to underlying FUS mutations. Furthermore, by using both GFP-tagged and untagged isogenic lines, there is currently no evidence that the GFP-tagging biased the observed phenotypes.

We show a stepwise acquirement of degenerative phenotypes starting with impaired distal axonal organelle trafficking followed by distal axon degeneration and finally motor neuron cell death (Figs. 2 and 4). While post mortem analysis revealed clear motor neuron loss in the spinal cord of the advanced FUS-ALS cases studied (Fig. 3a, b), the in vitro data showed only minor signs of neuronal cell body degeneration at time points of severe distal axon degeneration (Fig. 2g–i), suggesting a dying-back process as typically seen in a distal axonopathy[37]. hiPSC-derived motor neuron cultures are the only way to monitor early disease course in human and could explain the differences with the post mortem results. However, while DNA damage induction always led to distal axonal phenotypes and FUS cytoplasmic mislocalization, this was not seen when inducing a solely axonal degeneration by nocodazole treatment or ATP depletion

(Fig. 6m–o). This suggests that FUS-ALS is a neuronopathy rather than a pure distal axonopathy, but one with prominent and especially early distal axonal degeneration.

One of the most intriguing questions of age-related neurodegenerative diseases is how a somatic mutation causes neurodegeneration specifically in aged individuals. Evidently, DNA damage accumulates during aging (for review see ref. [30]). Of note, age-related motor neuron degeneration has been observed in mice lacking the DNA repair protein ERCC excision repair 1, endonuclease non-catalytic subunit[38,39]. However, whether mutation severity, age of onset, amount of DNA DSBs, appearance of neurodegenerative phenotypes and FUS aggregation do correlate remains to be shown in future studies.

## Methods

**Patient characteristics**. We included cell lines carrying "benign" (R521C, L) and "malign" (R495QfsX527, P525L) FUS mutations and systematically compared them to three control iPSC lines from healthy volunteers and to isogenic lines (Table 1). All procedures were in accordance with the Helsinki convention and approved by the Ethical Committee of the Technische Universität Dresden (EK45022009, EK393122012) and by the ethical committee of the University of Ulm (Nr. 0148/2009) and patients and controls gave their written consent prior to skin biopsy.

**Generation and expansion of iPSC lines**. Fibroblast or keratinocytes lines were established from skin biopsies or hair follicle cells obtained from familial ALS patients and healthy controls[28]. The generation and characterization of control iPSC lines was reported previously[28,40]. Fibroblast lines were reprogrammed as previously described[28,40,41]. Briefly, patient fibroblasts were reprogrammed using pMX-based retroviral vectors encoding the human cDNAs of *OCT4*, *SOX2*, *KLF4* and *cMYC* (pMX vectors). Vectors were co-transfected with packaging-defective helper plasmids into 293T cells using Fugene 6 transfection reagent (Roche). Fibroblasts were plated at a density of 50,000 cells/well on 0.1% gelatin-coated 6-well plates and infected three times with a viral cocktail containing vectors expressing *OCT4:SOX2:KLF4:cMYC* in a 2:1:1:1 ratio in the presence of 6 μg/ml protamine sulfate (Sigma Aldrich) and 5 ng/ml FGF2 (Peprotech). Infected fibroblasts were plated onto mitomycin C (MMC, Tocris) inactivated CF-1 mouse embryonic fibroblasts (in-lab preparation) at a density of 900 cells/cm² in fibroblast media. The next day media was exchanged to ES medium containing 78% Knock-out DMEM, 20% Knock-out serum replacement, 1% non-essential amino acids, 1% penicillin/streptomycin/glutamine and 50 μM β-Mercaptoethanol (all from Invitrogen) supplemented with 5 ng/ml FGF2 and 1 mM valproic acid (Sigma Aldrich). Media was changed every day to the same conditions. iPSC-like clusters started to appear at day 7 post infection, were manually picked 14 days post-infection and plated onto CF-1 feeder cells in regular ES-Media containing 5 ng/ml FGF2. Stable clones were routinely passaged onto MMC-treated CF-1 feeder cells (Globalstem) using 1 mg/ml collagenase type IV (Invitrogen) and addition of 10 μM Y-27632 (Ascent Scientific) for the first 48 h after passaging. Media change with addition of fresh FGF2 was performed every day.

iPSC lines from human hair keratinocytes were generated as described in refs. [42–44] by a lentivirus containing a polycistronic expression cassette encoding for Oct4, Sox2, Klf4, and c-Myc PMID19096035 produced in 70% confluent 10 cm dishes with Lenti-X 293T cells (Clontech, Mountain View, CA) by cotransfection of

**Fig. 7** Impairment of nucleo-cytoplasmatic FUS shuttling leads to DNA damage, neurodegeneration and aggregate formation. **a** Time course of nuclear FUS withdrawal after addition of PARP1 inhib. (top gallery) or etoposide (eto, bottom gallery) in WT FUS-GFP MNs at 21 DIV. **b** Distinct inhibitory kinetics of FUS-dependent DNA repair through etoposide and PARP1 inhib: recruitment-withdrawal of FUS-GFP to nuclear laser cuts as for Fig. 6a on WT FUS-GFP MNs after PARP1 inhibitor (top) or etoposide (bottom) addition at indicated time points, i.e. before and after loss of nuclear FUS (corresponding red boxes in **a**), bars: 10 μm (Supplementary Movie 12). **c** Quantification of **b**. **d** FUS-GFP recruited to Laser cut in MNs double-treated 24 h with compounds as indicated (Supplementary Movie 13). Note the inhibition of recruitment to the cut (boxed areas) in mutant FUS-GFP P525L cells (far left) through double treatment with DNA-PKi and PARP1i whereas DNA-PKi alone led to a rescue (Fig. 6c, Supplementary Movie 6), suggesting that PARP1 functions upstream of DNA-PK in DNA damage response. Furthermore, note that inhibiton of recruitment through etoposide or arsenite in WT FUS-GFP (Fig. 6c, Supplementary Movie 6) was reverted through double treatment with either DNA-PKi or AdOx, suggesting a potent counterbalancing of etoposide/arsenite-driven displacement of FUS from the nucleus, thereby rescuing FUS recruitment to the cut (boxed areas). **e** Quantification of **d**. **f** Compounds impacted on nuclear FUS-GFP levels in MNs: double treatments Mut versus WT as for **d**. Note how DNA-PK inhib. or AdOx drove FUS-GFP back into the nucleus under conditions that normally lead to nuclear export of FUS (i.e. PARP1, etoposide or arsenite treatment, Fig. 6e). **g** Western blot (WB) analysis of markers for c-NHEJ DNA damage response (phospho and total DNA-PK, KU70 & KU80) and a-NHEJ (LIG1, PARP1 total and cleaved) on total lysates of MNs treated for 72 h as indicated. GFP, β-actin served as loading controls. **h** Densiometric quantification of **g**, WT (blue) versus FUS (red), mock only. Graphs show fold change over mean WT (mock baseline = 1) to reveal phenotypic differences of quasi untreated FUS over WT. **i** Densiometric quantification of **g** for all treatments. Graphs show fold change over respective cell line mock to reveal the compound response of each line (WT, blue, versus FUS, red) over its respective mock baseline (= 1). Statistics (**h**, **i**): data are plotted as mean, error bars = STDEV, one-way ANOVA with post-hoc Bonferroni test (*, **, ***P values of 0.05, 0.01, and 0.001, respectively, N = 4)

the polycistronic vector (8 mg), the pMD2 vector (2 mg), and the psPAX2 (5.5 mg) vector (both Addgene, Cambridge, MA) using 100 mL of the PolyFect transfection reagent (Qiagen, Hilden, Germany; www.qiagen.com). Viral supernatant was collected at 48 and 96 h after transfection, concentrated using the Lenti-X Concentrator Kit (Clontech), resuspended in EpiLife medium, and stored in aliquots at −80 °C. For infection, up to $3 \times 10^5$ keratinocytes per well of a 6-well plate were infected with $5 \times 10^7$ proviral genome copies in EpiLife medium containing 8 mg/mL Polybrene (Sigma- Aldrich) at 2 sequent days. After another 24 h cells were detached using TrypLE Express (Invitrogen) and distributed onto 3 wells of a 6-well plate with already attached irradiated feeder cells ($3 \times 10^5$ cells per well irradiated with 30 Gy) in hiPS cell medium. Further on the cells were cultured in a 5% $O_2$ incubator. The medium was changed daily until arising colonies were large enough for mechanical passaging at about 2–3 weeks after transduction. Colonies displaying a clear stem cell morphology were picked and transferred onto irradiated MEFs or Matrigel-coated plates for further passage.

Stable clones were analyzed by qRT-PCR for silencing of viral transgenes prior to further experimental procedures.

**CRISPR/Cas9 genome editing.** To generate the two isogenic cell lines FUS WT-GFP and P525L-GFP, a patient-derived iPSC clone from a female patient carrying a heterozygous FUS R521C mutation was used (Table 1). The patient-specific *FUS* mutation was corrected at its mutation site and simultaneously C-terminal tagged with EGFP by CRISPR/Cas9-mediated genome editing via homology directed repair (HDR)[45]. For that a CRISPR/Cas9D10A vector (pX335B) containing the D10A mutant nickase version of Cas9 (Cas9n) and a pair of guide RNAs (gRNA) were used to create a double strand break (DSB) at the target site. The pX335B vector, containing Cas9n, was provided by the laboratory of Dr. Boris Greber, (Max Planck Institute for Molecular Biomedicine, 48149 Münster, Germany). As a HDR template a plasmid (pEX-K4) containing the FUS correction sequence (WT or P525L) plus EGFP-Tag was used (Eurofins Genomics).

To target the FUS R521C mutation site, two gRNA cassettes were designed. The gRNAs were manually selected by screening of the coding strand for suitable gRNA target sequences upstream of R521C mutation. DNA motifs screened for were CCN(N)19C or CCN(N)20C for target one (T1) and G(N)19NGG or G(N)20NGG for target two (T2). Target specific gRNAs (T1- gcgagtatcttatctcaagt; T2- gttaggtaggaggggcagat) then were cloned into the pX335B vector in two cloning steps. Successfully cloned vectors were identified via colony PCR after each ligation step. Positive clones were amplified, sequenced and used for transfection. The FUS correction sequences were designed according to the WT sequence of FUS (NCBI Ref. Seq. NC_000016.10). Homology arms covering the mutation site, and in a size of 500 bp upstream and 400 bp downstream of the induced DSB were used. The EGFP-Tag was added to the last exon of FUS with a 9 bp linker-DNA sequence. The sequences for WT-EGFP and P525L-EGFP were synthesized de novo and cloned into the pEX-K4 from Eurofins Genomics.

For gene targeting of human iPSCs, feeder-free iPSCs cultured in TeSR-E8 medium (Stemcell Technologies) were co-transfected with pX335B vector (containing the Cas9n cassette, two gRNAs and a puromycin selection cassette) and pEX-K4 vector (containing the FUS WT-EGFP or P525L-EGFP sequence) using FuGENE HD transfection reagent (Promega). 24 h after transfection, cells containing the pX335B vector were selected by treatment with 0.4 μg/μl Puromycin (InvivoGen) for another 24 h. After selection, cells were allowed to recover for 3–7 days and then passaged onto a new plate (2000 cells per 6-well). After 10–14 days EGFP-positive clones were picked and further cultured for characterization.

To identify successful homologous recombined clones, DNA was isolated (DNeasy Blood&Tissue Kit, Qiagen) from these cells and PCR was performed to confirm the presence of the EGFP-Tag within the genomic sequence of *FUS*. For this a forward primer (AGTTACCAGCCTCTCCAAGC) targeting FUS upstream of the used homology arms, and a reverse primer (CGGTGGTGCAGATGAACTT) targeting EGFP were used. After successful PCR the genotype of clones positive for FUS-EGFP was determined by PCR and sequencing. For the genotyping PCR, the forward primer (CAGTTGAACAGAGGCCATAGG) and reverse primer (CAGTTGAACAGAGGCCATAGG) were used targeting FUS up- and downstream of EGFP, including the mutation site. Amplification results in two PCR products (1257 bp for FUS with EGFP and 528 bp for FUS without EGFP), if cells are heterozygous for the introduced modification. Additionally, gDNA of non-transfected cells was amplified as negative control, resulting in the amplification of only one PCR product (528 bp for FUS without EGFP). To confirm the genotype by sequencing, PCR products were analyzed on a 1% agarose gel and the two different sized bands were purified from the gel (QIAquick Gel Extraction Kit, Qiagen). In order to identify if correction of FUS occurred on the originally mutated allele and not on the WT allele, both purified PCR products were sequenced (Supplementary Fig. 1d, e). The PCR product containing FUS with EGFP (Allele A) was sequenced using the reverse primer (CGGTGGTGCAGATGAACTT). The smaller PCR product containing FUS without EGFP (Allele B) was sequenced using the reverse primer (TGGGTGATCAGGAATTGGAAGG). The original genotype of the used patient-derived iPSC line was FUS-R521C/FUS-WT. The genotype of successfully modified clones WT-GFP and P525L-GFP is FUS-WT-EGFP/FUS-WT and FUS-P525L-EGFP/FUS-WT, respectively (Supplementary Fig. 1d, e).

To check for possible off-target effects of Cas9n both target specific gRNAs (T1 and T2) were checked via the "CRISPR Design" online tool from Zhang laboratory (http://crispr.mit.edu) and the "Off-spotter" tool from Pliatsika et al.[46]. The online tools confirmed that there are no genomic off-targets that are targeted from both gRNAs in combination, which could result in the introduction of a DSB at any untargeted site.

**Mycoplasma testing.** We checked every new cell line when entering the lab and after reprogramming, afterwards routinely check for mycoplasma every three to six months. We used the Mycoplasma Detection kits for conventional PCR according to manufacturer's instructions (Venor GeM, No 11–1025).

**In vitro differentiation of embryoid bodies.** iPSC colonies were grown under standard conditions, cleaned and treated with collagenase type IV (2 mg/ml, Invitrogen). Floating aggregates were collected and transferred into ultra-low attachment plates (NUNC) in regular ES-Media containing 5 μM Y-27632 (Ascent Scientific) for meso-/endodermal differentiation or ES-Media containing 5 μM Y-27632, 10 μM SB431542 (Tocris) and 1 μM Dorsomorphin (Tocris) for ectodermal differentiation. Two days later the medium was changed to the same conditions leaving out the Y-27632. After four days of EB formation, aggregates were plated onto gelatin (0.1%, Millipore) coated wells for meso-/endodermal differentiation or onto plates coated with Matrigel™ (BD Bioscience) for ectodermal differentiation. EBs were differentiated for two weeks using 77.9% DMEM (high glucose, Invitrogen), 20% FCS (PAA), 1% non-essential amino acids (Invitrogen), 1% penicillin/streptomycin/glutamine (Invitrogen) and 0.1% β-Mercaptoethanol (Invitrogen) for the meso-/endodermal lineage and 50% DMEM/F12 (Invitrogen), 50% Neurobasal (Invitrogen) containing 1:200 $N_2$ supplement (Invitrogen), 1:100 B27 supplement without vitamin A (Invitrogen), 1% penicillin/streptomycin/glutamine, 0.1% β-Mercaptoethanol and 1:500 BSA Fraction V (Invitrogen) for ectodermal differentiation.

**AP staining and immunofluorescence on iPSC colonies.** For pluripotency marker stainings, iPSC colonies were passaged as described above and grown on Matrigel™-coated coverslips in ES medium containing, 50% MEF-conditioned media (own preparation) supplemented with 5 ng/ml FGF2 (Sigma). Colonies were then stained for alkaline phosphatase according to the manufacturer's protocol (Millipore) or were fixed with 4% paraformaldhyde (PFA) in PBS for 10 min at RT for analysis of pluripotency markers by immunofluorescence. Fixed colonies were incubated for 2 h in blocking solution (3% normal horse serum and 0.05–0.2% Triton-X100 in PBS). Plates were incubated over night at 4 °C using the following primary antibodies: rabbit anti-Nanog (1:500), rabbit anti-Oct4 (1:1000), mouse anti-SSEA4 (1:500), mouse anti TRA-1-60 (1:500) (all from Abcam) and mouse anti-Sox2 (1:500, R&D Systems). Differentiated EBs were stained with rabbit anti-α-SMA (1:500, Sigma Aldrich), mouse anti-α-Fetoprotein (1:500, Abcam), rabbit anti-GATA4 (1:500, Abcam), and mouse anti-TUJ1 (1:1000, Covance), mouse anti-Actinin (Sigma, 1:200) and mouse anti Beta-Catenin (BD Bioscience 1:500).

**Karyotyping.** FUS iPSC and control cell lines were karyotyped using the HumanCytoSNP-12v array. All clones showing pathological SNPs were excluded (data not shown).

**Genotyping.** FUS iPSC lines were genotyped after all other characterization had been finished. This was done by a diagnostic human genetic laboratory (CEGAT, Tübingen, Germany) using diagnostic standards.

**Differentiation of human NPCs to spinal MNs.** The generation of human NPCs and MNs was accomplished following the protocol from Reinhardt et al.[26].

In brief, colonies of iPSCs were collected and stem cell medium, containing 10 μM SB-431542, 1 μM Dorsomorphin, 3 μM CHIR 99021 and 0.5 μM pumorphamine (PMA), was added . After 2 days hESC medium was replaced with N2B27 consisting of the aforementioned factors and DMEM-F12/Neurobasal 50:50 with 1:200 $N_2$ Supplement, 1:100 B27 lacking Vitamin A and 1% penicillin / streptomycin / glutamine. On day 4 150 μM ascorbic acid was added while Dorsomorphin and SB-431542 were withdrawn. 2 Days later the EBs were mechanically separated and replated on Matrigel coated dishes. For this purpose Matrigel was diluted (1:100) in DMEM-F12 and kept on the dishes over night at room temperature. Possessing a ventralized and caudalized character the arising so called small molecule NPCs (smNPC) formed homogenous colonies during the course of further cultivation. It was necessary to split them at a ratio of 1:10–1:20 once a week using Accutase for 10 min at 37 °C.

Final MN differentiation was induced by treatment with 1 μM PMA in N2B27 exclusively. After 2 days 1 μM retinoic acid (RA) was added. On day 9 another split step was performed to seed them on a desired cell culture system. Furthermore the medium was modified to induce neural maturation. For this purpose the developing neurons were treated with N2B27 containing 10 ng/μl BDNF, 500 μM dbcAMP and 10 ng/μl GDNF. Following this protocol it was possible to keep the cells in culture for over 2 months.

**Treatments and inhibitors**. Adenosine-2′,3′-dialdehyde (AdOX, arginine methyltransferase inhibitor, Sigma-Aldrich A7154) was dissolved in water and traces of HCl to obtain a 10 mM stock, DNA-PK inhibitor NU7441 (Tocris Cat. No. 3712) was dissolved in DMSO to obtain a 5 mM stock, Sodium Arsenite (Sigma-Aldrich S7400) was dissolved in water to obtain a 5 mM stock, Etoposide (Sigma-Aldrich E1383) was dissolved in DMSO to obtain a 500 µM stock, Gallo-tannin (PARG inhibitor, Santa Cruz Biotechnology sc-202619) was dissolved in water to obtain a 30 mM stock and ABT-888 (PARP1 inhibitor, Santa Cruz Biotechnology sc-202901) was dissolved in DMSO to obtain a 20 mg/ml stock. For treatment of hiPSC-derived spinal MNs in Xona Microfluidic chambers (see below), compounds were added either exclusively to the proximal or distal site for locally restricted application in compartmentalized cultures, thereby allowing to distinguish between local versus remote compound action at the distal and proximal readout positions (refer also to 'Life cell imaging' below). Cells were incubated for 72 h before live imaging and final concentrations were as follows: 10 µM for AdOx, 2 µM for DNA-PK, 30 µM for PARG inhibitor, 2 µg/ml for PARP1 inhibitor and 2 µM for Etoposide.

For uncompartmentalized cell cultures, all compounds were added only 24 h before imaging (e.g. for the Laser cutter experiments, if not otherwise stated in time course experiments) with final concentrations as follows: 5 µM for AdOx, 5 µM for DNA-PK, 300 µM for PARG inhibitor, 20 µg/ml for PARP1 inhibitor, 5 µM Sodium Arsenite and 5 µM for Etoposide. DMSO was used for Mock controls.

**Microfluidic chambers**. The MFCs were purchased from Xona (RD900). At first, Nunc glass bottom dishes with an inner diameter of 27 mm were coated with Poly-L-Ornithine (Sigma-Aldrich P4957, 0.01% stock diluted 1:3 in PBS) overnight at 37 °C. After 3 steps of washing with sterile water, they were kept under the sterile hood for air drying. MFCs were sterilized with 70% Ethanol and also left drying. Next, the MFCs were dropped onto the dishes and carefully pressed on the glass surface for firm adherence. The system was then perfused with Laminin (Roche 11243217001, 0.5 mg/ml stock diluted 1:50 in PBS) for 3 h at 37 °C. For seeding cells, the system was once washed with medium and then 10 µl containing a high concentration of cells ($3 \times 10^7$ cells/ml) were directly injected into the main channel connecting two wells. After allowing for cell attachment over 30–60 min in the incubator, the still empty wells were filled up with maturation medium. This method had the advantage of increasing the density of neurons in direct juxta-position to microchannel entries whereas the wells remained cell-free, thereby reducing the medium turnover to a minimum. To avoid drying out, PBS was added around the MFCs. Two days after seeding, the medium was replaced in a manner which gave the neurons a guidance cue for growing through the microchannels. Specifically, a growth factor gradient was established by adding 100 µl N2B27 with 500 µM dbcAMP to the proximal seeding site and 200 µl N2B27 with 500 µM dbcAMP, 10 ng/µl BNDF, 10 ng/µl GDNF and 100 ng/µl NGF to the distal exit site. The medium was replaced in this manner every third day. After 7 days, the first axons began spreading out at the exit site and cells were typically maintained for up to six weeks.

**Immunofluorescence stainings**. For immunofluorescence staining, cells were washed twice with PBS without $Ca^{2+}/Mg^{2+}$ (LifeTechnologies) and fixed with 4% PFA in PBS for 10 min at room temperature. PFA was aspirated and cells were washed three times with PBS. Fixed cells were first permeabilized for 10 min in 0.2% Triton X solution and subsequently incubated for 1 h at RT in blocking solution (1% BSA, 5% donkey serum, 0.3 M glycine and 0.02% Triton X in PBS). Following blocking, primary antibodies were diluted in blocking solution and cells were incubated with primary antibody solution overnight at 4 °C except for the γH2A.X antibody which was kept for only 2 h at room temperature on the fixed material. The following primary antibodies were used: chicken anti-SMI32 (1:10,000, Covance), mouse anti-FUS (1:5000, Sigma-Aldrich), rat anti-meFUS (1:100[5],), rabbit anti-beta-III-Tubulin (1:3000, Covance), mouse anti-Hb9 (1:100 Development studies Hybridoma Bank), rabbit anti-Islet (1:500, Abcam), mouse anti-γH2A.X (1:500 Millipore), rabbit anti-ChAt (1:500, Chemicon), rabbit anti-53BP1 (1:1000, Novusbio). Nuclei were counter stained using Hoechst (LifeTechnologies).

**Western blotting**. Western blot analysis (FUS) was performed as described in Prause et al.[47]. DNA damage pathway was analyzed as follows. Lysates of neuronal cell cultures were prepared as described[48] with modified RIPA buffer consisting of 50 mM Tris-HCl (pH 7.4), 1% Nonidet-P40, 0.25% sodium deoxycholate, 150 mM NaCl, 1 mM EDTA, 1 mM NaVO4, 2 mM NaF (all Sigma-Aldrich), Complete protease inhibitor cocktail (Roche). Protein amount was measured by BCA assay (Thermo Fisher Scientific). After SDS–PAGE and transfer of proteins onto nitro-cellulose membranes (GE Healthcare), specific proteins were detected using the indicated primary antibodies and horseradish peroxidase-conjugated donkey anti-rabbit and sheep anti-mouse antibodies (GE Healthcare). Detection of proteins on X-ray films (GE Healthcare) was accomplished with enhanced chemiluminescent reagent (Amersham). Antibodies were purchased as indicated: DNA-PK (#4602), PARP1 (#9542), KU80 (#2180, Cell Signaling Technology), β-actin (A5441, Sigma), phospho-DNA-PK S2056 (ab18192), phospho-DNA-PK S2056 (ab18192, Abcam), LIG1 (ab177946), KU70 (ab3114), GFP (ab290, Abcam).

**Live cell imaging and tracking analyses**. For tracking of lysosomes and mito-chondria, cells were double-stained live with 50 nM Lysotracker Red DND-99 (Molecular Probes Cat. No. L-7528) and 50 nM Mitotracker Deep Red FM (Molecular Probes Cat. No. M22426). For measuring mitochondrial membrane potential (and tracking as well), cells were stained with 200 nM Mitotracker JC-1 (Molecular Probes Cat. No. M34152). Trackers were added directly to culture supernatants and incubated for 1 h at 37 °C. Imaging was then performed without further washing of cells. Live imaging of compartmentalized axons in Xona Microfluidic Chambers (MFC) was performed with a Leica HC PL APO 100 × 1.46 oil immersion objective on an inversed fluorescent Leica DMI6000 microscope enclosed in an incubator chamber (37 °C, 5% CO2, humid air) and fitted with a 12-bit Andor iXON 897 EMCCD camera (512×512, 16 µm pixels, 229.55 nm/pixel at 100X magnification). For more details, refer to https://www.biodip.de/wiki/Bioz06_-_Leica_AFLX6000_TIRF. Excitation was performed with a TIRF Laser module in epifluorescence (widefield) mode with lines at 488, 561 and 633 nm. Fast dual color movies were recorded at 3.3 frames per second (fps) per channel over 2 min (400 frames in total per channel) with 115 ms exposure time as follows: Lysotracker Red (excitation: 561 nm, emission filter TRITC 605/65 nm) and Mitotracker Deep Red (excitation: 633 nm, emission filter Cy5 720/60 nm) or for Mitotracker JC-1 with excitation at 488 nm and fast switching between emission filter FITC 527/30 nm (green channel for compromised membrane potential) and TRITC 605/65 nm (red channel for intact membrane potential). Movie acquisition was performed at strictly standardized readout positions within the micro channels of the micro groove barrier that separated the proximal seeding site from the distal axonal exit as illustrated in Fig. 2a. Specifically, the readout windows were located either just adjacent to the channel exit (distal readout) or the channel entry (proximal readout).

**Tracking analysis**. Movies were analyzed with FIJI software using the TrackMate v2.7.4 plugin for object (lysosomes and mitochondria) recognition and tracking. Settings were as follows: pixel width: 0.23 µm, pixel height: 0.23 µm, voxel depth: 1 µm, crop settings: not applied, select a detector: DoG detector with estimated blob size: 1.6 µm, threshold: 45, median filter: no, subpixel localization: yes, initial thresholding: none, select view: HyperStack Displayer, set filters on spots: quality above 45, select a tracker: linear motion LAP tracker, initial search radius: 2 µm, search radius: 2 µm, Max. frame gap: 2, set filters on tracks: track duration ≥3 s. Typically, 200–500 tracks per movie were obtained and analyzed with respect to track displacement (measure for processive, i.e. straight, motility as opposed to undirected random walks) and mean speed. Tracking is illustrated in Supplementary Movie 5. Results were assembled and post-filtered (threshold for track displacement ≥1.2 µm) in KNIME and MS Excel and bulk statistics analyzed and displayed as box plots in GraphPad Prism 5 software (Figs. 3b, c, e–i, 4i–m and 5f–j). Box plot statistics are provided in Supplementary Tables 1–17. Box plot settings: whiskers from 1–99%, outliers as dots, boxes from 25–75 percentile, median as horizontal center line, mean as cross. Significant differences between conditions (i.e. cell lines, compound treatments, etc.) were revealed with the nonparametric Kruskal-Wallis test for non-Gaussian distributions and Dunns post hoc test with significance level $P \leq 0.05$ and 95% confidence interval. Box plots represent batch results merged from all apparently healthy control lines (Ctrl 1–3) or mutant ALS-FUS lines (R521C, R521L, R495QsfX527) and three independent experiments (i.e. differentiations, refer to Fig. 1a). A minimum of 5 movies (showing 2 micro channels each) was acquired at each readout positions (distal versus proximal) per line, condition and experiment resulting in a minimum of 15 movies in total for the batch analysis. We confirmed that all control and FUS lines were phenotypically indistinguishable, thereby validating our batch analysis (Supplementary Figs. 2–9).

**Static analysis of cell organelles**. For analysis of organelle count and morphology (mitochondria: elongation; lysosomes: diameter), object segmentation, thresholding and shape analysis was performed with a sequence of commands in FIJI software executed with Macro1 for mitochondria:

```
run("Slice Keeper", "first=1 last=1 increment=1");
run("Grays");
run("Subtract Background…", "rolling=3");
setAutoThreshold("IsoData dark");
//run("Threshold…");
run("Convert to Mask");
run("Set Measurements…", "area fit shape feret's redirect=None decimal=5");
run("Analyze Particles…", "size=4-Infinity pixel
circularity=0.00–1.00 show=Ellipses display summarize");
```

and Marco2 for lysosomes:

```
run("Slice Keeper", "first=1 last=1 increment=1");
run("Grays");
run("Enhance Contrast…", "saturated=0.1 normalize");
run("Subtract Background…", "rolling=5");
setAutoThreshold("Yen dark");
run("Convert to Mask");
run("Set Measurements…", "area fit shape feret's redirect=None decimal=5");
run("Analyze Particles…", "size=3-Infinity pixel
circularity=0.40–1.00 show=Ellipses display summarize");
```

These macros returned result tables containing the aspect ratio of fitted eclipses (long:short radius) that was taken as the measure for mitochondrial elongation as well as the outer Feret's diameter that was taken as lysosomal diameter. The same set of movies as for the tracking analysis (see above) was used (first frame only). Typically, hundreds of organelles were analyzed per movie. For bulk statistics, the same batch analysis as for the tracking analysis was performed with resultant distributions displayed as box plots (Figs. 3h, i, 4l, m and 5i, j).

For analysis of mitochondrial membrane potential (ratio JC-1 red:green channel), object segmentation was performed with the channel of higher intensity (mostly the red) to generate a selection limited to mitochondria using Macro3:

    resetMinAndMax();
    title=getTitle();
    run("Slice Keeper", "first=1 last=1 increment=1");
    run("Subtract Background…", "rolling=10");
    setAutoThreshold("Default dark");
    //run("Threshold…");
    run("Convert to Mask");
    run("Create Selection");

The resultant selection was saved as region of interest (ROI) and applied on both channels to reveal the total integral intensity and area of mitochondria and background in both channels using the "Measure" command. After area normalization and background subtraction, ratios of integral intensity red:green were taken as mean membrane potential per movie (first frame only) and batch-analyzed as for the tracking analysis (see above). The resultant distributions were displayed as box plots on a log scale (Figs. 3g, 4k and 5h).

**DNA damage laser cutting assay**. The UV lasercutter setup utilized a passively Q-switched solid-state 355 nm UV-A laser (Powerchip, Teem Photonics) with a pulse energy of 15 μJ at a repetition rate of 1 kHZ. With a pulse length of <350 ps this resulted in a peak-power of 40 kW, of which typically less than 5% was used to cut tissues. The power was modulated using an acousto-optic modulator (AOM, AA.MQl l0-43-UV, Pegasus Optik). The laser-beam diameter was matched to the size of the back-aperture of the objectives by means of a variable zoom beam expander (Sill Optics). This enabled diffraction-limited focusing while maintaining high transmission for objectives with magnifications in the range of 20–100X. Point-scanning was realized with a pair of high-speed galvanometric mirrors (Lightning DS, Cambridge Technology). To this end, the scanning mirrors were imaged into the image-plane of the rear port of a conventional inverted microscope (Axio Observer Z1, Zeiss) with a telecentric f-theta objective (Jenoptik). In order to ease adjusting parfocality between the cutter and the spinning disk and to compensate for the offset between the positions of the back-planes of different objectives, the scan-mirrors and the scan-optics were mounted on a common translation stage. In the microscope reflector cube, a dichroic mirror reflected the UV light onto the sample and transmitted the fluorescence excitation and emission light. A UV-blocking filter in the emission path protects the camera and enables simultaneous imaging and ablating. The AOM, the galvanometric mirrors as well as a motorized stage (MS 2000, ASI) with a piezo-electric actuator, on which the sample is mounted, were computer-controlled using custom-built software (LabView, National Instruments) enabling cutting in 3D. Diffraction-limited cutting with little geometric distortion, high homogeneity of the intensity and good field flatness was possible in the entire field of view of the spinning disk. The maximum depth is limited by the free working distance of the objective used and the travel of the piezo-actuator (100 μm). A Zeiss alpha Plan-Fluar 100 × 1.45 oil immersion objective was used and 24 laser shots in 0.5 μm-steps were administered over a 12 μm linear cut.

**Imaging and quantification of cytosolic FUS aggregation in FUS-GFP tagged lines**. Untreated, Mock- or compound-treated cells were fixed and a Z-stack of 20 images in 0.5 μm-steps was acquired with standard filter settings for GFP fluorescence. The Z-stack covered the full range from the bottom cytosolic to the top nuclear focal plane to capture all FUS protein. For quantification of resultant maximum intensity projections, the total integral GFP intensity in the nucleus, total cytosol as well as in cytosolic aggregates only was analyzed with FIJI software.

**Quantification and statistics**. Randomly assigned images of different experiments were quantified on day 14 of neuronal differentiation to evaluate MN differentiation capacity. To analyze DNA damage, images of NPCs and of mature MN 30 days after differentiation initiation were examined and the mean number of γH2AX foci representing DSBs per nucleus was determined.

A minimum of three independent experiments based on three different differentiation pipelines were always performed. Statistical analysis was performed using GraphPad Prism version 5.0. If not otherwise stated, one-way ANOVA was used for all experiments with post-hoc Bonferroni post test to determine statistical differences between groups. *$P < 0.05$, **$P < 0.01$, ***$P < 0.001$, ****$P < 0.0001$ were considered significant. Data values represent mean ± SDTEV unless indicated otherwise.

**Electrophysiology**. Patch-clamp recordings were performed as described previously[28]. To perform electrophysiological experiments during week 7 of total differentiation, we seeded 300,000 cells per Matrigel-coated coverslip in a 24 well plate on day 25. To ensure recording of MNs we selected large (>20 pF) multipolar neurons only. Furthermore, the internal patch solution was filled with secondary antibody alexa 488 to allow MN identification after an additional immunostaining step using alexa 555 against SMI32 primary antibody.

**Histology**. Human post mortem samples (spinal cords, $n = 3$ of FUS mutations (NLS mutation) and 4 for controls), were obtained from the Amsterdam Academic Medical Center (AMC), Division of Neuropathology, Department of Pathology ALS Bank following the guidelines of the local ethics committee. The spinal cords of these clinically confirmed FUS-ALS patients, as well as age-matched controls had been obtained within 6–12 h after death. Tissues were used in compliance with the Declaration of Helsinki. All FUS-ALS patients suffered from clinical signs and symptoms of lower and upper MN disease with the eventual involvement of Cortex, brain stem motor nuclei. Age-matched control patients did not show any neuropathological anomalies. Transverse paraffin sections (3–4 μm in thickness) of human (lumbar, thoracic, cervical) spinal cord were cut on a microtome. Sections were placed on silane-coated nuclear slides, de-waxed, rehydrated and heated in citrate buffer for antigen retrieval. Processed sections were incubated with primary antibodies (Mouse anti H2AX (Millipore), rabbit anti-FUS- (NOVUS)), each (1:100) for 1 h at room temperature. Appropriate HRP secondary antibodies were used (1:200, Vector Laboratories, USA) for 1 h, followed by DAB visualization (DAKO, Denmark). For immunofluorescence secondary antibodies conjugated with Alexa fluorophore (Invitrogen) were used. Staining patterns were visualized using a Zeiss LSM 700 confocal microscope. The resulting confocal images were processed using the Zeiss LSM software and Adobe Photoshop CS5. DAB immunohistochemical sections were photographed using an Axioplan microscope (Zeiss) with an Axio Cam HR camera using 63X oil immersion lens (Zeiss).

**Data availability**. All data related to the manuscript is included in the main text or Supplementary Files, or available from the authors upon reasonable request.

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

## Acknowledgements

We acknowledge the great help in cell culture by Sylvia Kanzler, Anett Böhme and Katja Zoschke. Ronny Sczech helped with the tracking analysis. The Light Microscopy Facility (LMF) of CMCB (Center for Molecular and Cellular Bioengineering,Technische Universität Dresden) provided invaluable support for all live imaging experiments. The meFUS antibody was provided by D. Dormann. We sincerely thank Stichting ALS Nederland and ALS Centre Netherlands (E. Aronica) for ALS research support. This work was supported, in part, by the Else Kröner foundation to M.N., "Deutsche Gesellschaft für Muskelerkrankungen (He2/2)" to A.H., the Roland Ernst Stiftung Saxony to A.H., the MeDDrive program of the Medical Faculty at the Technische Universität Dresden to A.H., BIOCREA GMBH to A.H., the NOMIS foundation to A.H., grant No 281903 of the European Research Council to S.W.G, the Helmholtz Virtual Institute "RNA dysmetabolism in ALS and FTD (VH-VI-510)" to A.H, T.M.B., A.C.L., S.L. and A. S., an unrestricted grant by a family of a deceased ALS patient to A.H, the German Motor Neuron Disease Network (BMBF-MND-Net; Funds 360644) to J.W., the Inter-disciplinary Centre for Clinical Research (IZKF Aachen, N7-4), the German Myopathy Society (DGM) and the Initiative Therapieforschung ALS e.V. to A.G. and J.W., the Frick Foundation for ALS research to E.S., the Minna-James-Heinemann-Stiftung to E.S., Association Française contre les Myopathies (AFM) to E.S., the Muscular Dystrophy Association (MDA), the EU Joint Programme − Neurodegenerative Disease Research (JPND; grant numbers ZonMW 733051075 (TransNeuro) and ZonMW 733051073 (LocalNMD)) and an ERC consolidator grant (ERC-2017-COG 770244) to E.S. and the Max Planck Society to E.S.

## Author contributions

M.N., A.P. and A.H. designed all of the experiments. T.M.B., A.C.L., S.L., I.P., M.F., E. S., X.L., P.R., J.S., J.H.W., A.F., K.H., A.Hy, A.S. and A.H. generated and provided the study material. J.J. generated isogenic lines. M.N., N.S., F.W. and S.P. performed and analyzed the functional assays. M.N., A.P., S.W.G., A.Hy and A.H. designed, performed and analyzed the laser cutter experiments. M.N., A.P., F.P.-M. and A.H. designed, performed and analyzed all live cell imaging. M.N., A.P., J.J., F. P.-M., N.S., F.W., S.P., A.V., R.G. M.J, N.C. and A.H. designed, performed and analyzed all cell culture experiments. A.G., D.T. and J.W. did the neuropathology analysis. A.H. supervised the project, M.N., A.P. and A.H. wrote the manuscript and all other authors critically revised the manuscript.

## Additional information

**Competing interests:** The authors declare no competing financial interests.

