## [Peer Review File · Nature Communications]

Reviewer #1 (Remarks to the Author):

Comments

Although this is an interesting paper and is well presented. Obvious strengths include the use of isogenic cell lines. However, I do have a number of major concerns with this paper:

- This paper focuses on FUS-related neurodegeneration and describes the disruption of the distal axon machinery, DNA repair and the time of cytoplasmic inclusions. However, there does not appear to be an actual read out for cell death/neurodegeneration in their iPSC experiments.
- The authors use the term distal axonopathy. However, they do not show that the axons are dying back, which would be the sine non-quo of axonopathy.
- Furthermore, just because they could identify issues with the distal axon (organelle hang-up), this does not necessarily imply that the problem is in the axon. The axon requires some serious cellular machinery in the cell body for its maintenance. Thus, I think a more likely explanation of their findings is that there is a problem in the cell body and that the consequence of this are then observed in the distal axon. This is especially true given the hypothesis that DNA repair is central to the process of FUS-induced neurodegeneration. Is there DNA repair ongoing in the axon? I suspect not, so it is not clear how one can say that the axon the site of the problem.

Minor points

Page 6: what is meant by "after extended maturation"? Please state the timeframe.

Page 3: FUS mutations account for less than 1% of sporadic disease, not 3% as stated.

Reviewer #2 (Remarks to the Author):

In this study, Naumann et al. aim to better characterize the initial events in the pathophysiology of FUS-ALS. By generating an isogenic patient iPSC-derived motor neuron model, the authors very elegantly tested the spatiotemporal link between DNA damage, FUS aggregate formation, and neurodegeneration. The major novelty of the study lies in the discovery that FUS-ALS patient mutations cause an increased DNA damage phenotype and distal axonal degeneration prior to FUS aggregation, and that restoring nuclear FUS recruitment to DNA damage sites can modulate the axonal phenotype.

Collectively, this study furthers our understanding of early phenotypes associates with FUS-ALS, providing some interesting insights into targets for therapeutic intervention. Based on these, I think that the manuscript is of strong potential interest for publication in Nature Communications. However, I also feel that some of the connections linking DNA damage to distal axonopathy need to be elaborated to support the model that the authors propose. In particular, while the authors show that treatments, such as inhibiting DNA-PK and PARG inhibition, prevent FUS accumulation in the cytoplasm, DNA-PK inhibition is also likely to inhibit DSB repair through NHEJ. A potential issue with these experiments in the manuscript is that DNA repair is assumed, but not directly measured, following these treatments. Conducting these experiments would help clarify whether it is DNA repair per se, or the cytoplasmic accumulation of FUS, that underlies the development of distal axonopathies. Below you will find a list of comments that I believe will further strengthen the manuscript:

Comments:

1. Are there differences in the maturity/electrophysiological properties of the mutant neurons versus control neurons? How do those measurements recorded for the mutant cells (Fig.1G-L) compare to control lines? What is mutant FUS cell line C10?
2. Figure 1F: in the text it is mentioned that cell death was unaffected, but how was this determined? Quantification of this phenotype would be useful. It is also very difficult to see the colocalization between methylated-FUS and FUS in the cytosolic aggregates. According to the model presented in the text, methylated-FUS should be primarily cytoplasmic in localization but by immunostaining it looks primarily nuclear?

3. Figure 1G: What time point were the electrophysiological recordings conducted at? 30d of differentiation?
4. "...and showed signs of polyubiquitination (Fig.1D)" is misleading since those experiments were performed in HeLa cells, but in the text it seems to refer to the inclusions observed in iPSC-derived FUS-ALS motor neurons.
5. The order of Fig. 2 and 3 should be changed, as the data presented in Fig. 3 is discussed first in the text.
6. Statistical analysis of data presented in Fig.3D-H, Fig.4E-I, Fig5.F-M is required.
7. Could the authors be clearer about the exact mutant line(s) analyzed throughout the paper? They mention generating iPSCs from 4 different patient lines but it appears as though the majority of data is a collection of data from three different mutant lines (i.e. Fig.3D-H). Were there any phenotypic differences observed between the different mutant lines? A more direct comparison (rather than the data presented separately as per Fig.S4-6) would be helpful. Also, it should be clarified that the FUS corrected data (Fig.3D-H) represents only the R521C corrected line (Fig.2D). How did the authors control for off-target effects of Cas9? How were the two alleles separately sequenced?
8. The phenotypes described in Fig.4 should also be characterized in the isogenic FUS corrected cells.
9. The finding that DNA DSB levels correlate with mutation/disease severity is very interesting, but the data is not presented in the manuscript other than the correlation values. Could the authors include that data? Additionally, the images corresponding to the data quantified in Fig.4B should be shown i.e. immunostaining for γ H2AX as well as an NPC marker (SOX2, Nestin, etc.).
10. The finding that etoposide-induced DNA damage caused a distal axon phenotype and not a proximal axonal abnormality (Fig.4) is very interesting. Could the authors comment on this phenomenon? Similarly, could the authors speculate on how restoring FUS recruitment to damage sites rescues the distal axonal phenotype?
11. PARG inhibition resulted in restored recruitment of mutant FUS to damage sites. Since we know that mutant FUS has an impaired interaction with other repair components i.e. HDAC1, it would be interesting to test whether restored recruitment of mutant FUS is sufficient to restore capacity for repair of damage sites via NHEJ.
12. The legend for Fig.5A states "mutant P525L FUS failed to repair DNA damage", but the experiment is testing recruitment to sites of DNA damage rather than repair per se. Additionally, the authors should perform immunostaining for a marker of DSBs i.e. γ H2AX to demonstrate the line of recruitment corresponds to actual DNA breaks. Along the same lines, pg.9 "We next asked whether interfering with nucleo-cytoplasmic shuttling of FUS is sufficient to impair DNA damage repair" is an inaccurate statement since capacity of the cells to repair damaged DNA is never measured. Rather, the experiments examine recruitment of FUS to sites of DNA damage.
13. Related to the above, the authors show that inhibition of either DNA-PK or PARG is sufficient to prevent the cytoplasmic accumulation of FUS and ameliorate axonal pathologies. However, DNA-PK activity is also essential for NHEJ. The authors should therefore measure DNA repair following treatment with DNA-PK and PARG inhibitors. These results should further be factored into the proposed model.
14. The observation that DNA-PK activity causes the export of FUS following DNA damage generation is potentially interesting, and suggests that in WT MNs, this might serve a physiological function. The authors should at least comment on this, and further clarify based on the experiments suggested in point 13, what happens to DNA repair efficiency when FUS export is inhibited.
15. Pg.13 "...we show here that FUS-ALS is caused by impairment of proper DNA damage response..." This is a bold statement that is not supported by the data presented in the manuscript. What is shown is that the distal axonopathy phenotype is caused by impaired DNA damage response signaling via abnormal FUS targeting.
16. Pg.14 "...treatment with PARG is sufficient to restore FUS recruitment to DNA damage sides...". Sides should be replaced with sites.

Reviewer #3 (Remarks to the Author):

Naumann et al. have investigated the relationship between DNA damage response, FUS cytoplasmic aggregation and a distal axonopathy observed in iPS-derived motor neurons carrying ALS-related FUS mutations. They propose that DNA damage is an event upstream of aggregate formation and axonopathy and may represent a novel therapeutic target.

The authors have generated iPS cells from ALS patients (R521C, R521L, R495QfsX527) and used CRISPR/Cas9n method to modify one line (R521C) into isogenic lines carrying WT-FUS-GFP or P525L-FUS-GFP. They determined that FUS mutant iPS-derived motor neurons accumulate cytoplasmic inclusion after 30 days of differentiation and convincingly demonstrate that mutant FUS motor neurons present a deficit of distal axon trafficking. Then, the authors show that treatment with Etoposide induces DNA double strand breaks (GammaH2AX staining) in control NPC and motor neurons and strikingly reproduces the distal axonopathy observed in mutant neurons. This phenotype as well as nuclear localization and recruitment of mutant FUS at DNA damage sites are rescued by treatment with PARG and DNA-PK inhibitors or a methyltransferase inhibitor (AdOx) previously shown to increase FUS nuclear import (Dormann 2012). Inversely, treatment of wild-type neurons with PARP inhibitor, etoposide or arsenite leads to cytoplasmic mislocalization, and PARP inhibitor induces abnormal axonal trafficking. Based on the observation that prolonged PAR activity by PARG inhibition restores mutant phenotypes, the authors propose that double strand DNA breaks lead to FUS mislocalization and neurodegeneration.

The link between FUS and the DNA repair machinery has been extensively studied (as appropriately mentioned throughout the manuscript). Indeed, previous studies showed that FUS is rapidly recruited to DNA damage sites (DDS) in a PAR-dependent manner (Mastrocola et al., 2013; Wang 2013; Rulten 2013) and the ALS-mutant FUS proteins are recruited to the DNA damage site but do not recruit effectively HDAC1 leading to an impairment of DNA damage repair (Wang et al., 2013; Qiu 2014). PARP inhibitors were previously shown to prevent recruitment of FUS to DNA damage (Rulten 2013; Mastrocola 2013; Patel 2015) and a link between DNA damage and FUS cytoplasmic translocation was already proposed (Deng 2014). Increased DNA damages recognized by GammaH2AX blots or immunostaining was also reported in transgenic mice (Qiu 2014) and autopsy samples (Deng 2014; Wang 2013). Hence, this study builds on several previous reports and confirms the link between FUS mislocalization and DNA damages. The novelty comes mostly from the system used: iPS-derived motor neurons, and isogenic cell lines expressing endogenous levels of wild-type and mutant GFP-FUS. The most important finding is the clear relationship between FUS mutation and abnormal distal axonal trafficking. Importantly, axonal deficit could be restored by targeting FUS nucleocytoplasmic trafficking (AdOX or DNA-PK inhibitor) and enhancing PAR activity (PARG inhibitor) which may have important implication for therapeutic intervention. While the authors propose that DNA damage is an upstream event, the model drawn in Fig6D rather shows that mislocalization preexists and is further enhanced by DNA damages that are less efficiently repaired in presence of FUS mutation. In this respect the title and abstract are misleading and the authors should rather stress the vicious circle enhanced by DNA damages. Overall, this study clearly demonstrates a central role for DNA damage repair in FUS pathology and the potential of targeting this pathway to restore FUS localization and function. In this reviewer's view the tools and findings from this study are of high relevance for the field. Several concerns mostly related to the presentation of the data should be addressed before consideration for publication in Nature Communication.

Major concerns:

1. While the introduction of the manuscript is well written, the authors should make additional efforts to describe the results obtained. It is often not clear from the text or legend which lines are being used (especially when figures are labeled only with "FUS" or "Ctrl" (Fig 1C-D and Fig 3)). The authors should clarify whether the graphs on Fig 3D-H represent combined results from all lines.

2. Figure 2 is not mentioned in the text, and it would be helpful to have a description of the different lines in the results instead of only in Table S1. Importantly, it is not clear from the text and Fig 2 which lines contain GFP, and whether the R521C line has been corrected to wild-type without GFP (apparently not from Table S1).
3. Similarly, it is not clear which line has been included in Fig 1C (only 2 controls, and mutant "P521L" and "P521C" that are not patient mutations).
4. The authors should clarify why Figure 1D has been performed in Hela cells, what does "ctrl and FUS" mean in this context, and why GFP antibodies recognize multiple bands below 100KD. Most importantly the authors claim that inclusions found in FUS-R521C mutant neurons (Fig 1F) "showed signs of polyubiquitinylation (Fig 1D)" while this has been done in Hela cells. They need to provide a similar blot from iPSC-derived neurons, and should also co-stain FUS inclusions with ubiquitin antibody.
5. The quality of the images is often not appropriate and results difficult to evaluate due to stainings that are too dim (Fig 1B, 1E, 1G, 2B). FUS inclusions are very difficult to see and the authors should provide higher magnification images (Fig 1F and 4E). It is also not possible to determine whether inclusions are labeled with meFUS antibody (they rather appear green than yellow). The authors should present separately all the channels of FUS, meFUS, Hoescht and SMI32 staining at 30 days differentiated motor neurons.
6. In legend of Figure 1J, 90.7% of motor neurons fired a single action potential and 32% fired repetitive action potential. These numbers should be clarified.
7. The authors should indicate the differentiation time of the iPSC-derived motor neurons in Figure 3, and determine whether Gamma-H2AX staining is abnormal in iPSC-derived neurons at 14 days of differentiation when they have not observed FUS-R521C mislocalization. It would also be important to clarify whether the time dependency reported in Fig 1 for R521C is also observed with more severe mutations.
8. Immunohistochemistry in patient samples is used to argue that a distal axonopathy rather than proximal axonopathy also occurs in vivo. However, images provided in Fig3 I-L are not convincing, with strong spinal motor neuron loss in I and lack of controls in J-K-L preventing the reader to determine whether the thinning of ventral roots is only moderate. In this reviewer's view this claim is neither supported by the results provided nor necessary to the study.
9. The authors claim at several places that DNA damage influence neurodegeneration but they do not provide any data about neuronal survival. They should replace neurodegeneration by axonopathy (including in the title).
10. The GammaH2AX in Fig 2A, J, K and L is confusing. The images provided in Fig2A after Etoposide treatment of control cells seem to show GammaH2AX staining in the cytoplasm rather than the nucleus. The quality and magnification of the images need to be improved. In addition the authors describe a cytoplasmic staining for GammaH2AX in patient autopsy samples (Fig 4J-L) which would be in contrast with previous reports (Wang 2013, Qiu 2014).
11. The authors should discuss the difference between their experiments showing that mutant GFP-P525L-FUS is not recruited to the DNA damage site (DDS) and results from Wang et al. (2013), Rulten et al. (2015) and Mastrocola et al. (2013) where overexpressed mutant R521C, R521G, or R524S-FUS are recruited to DDS.
12. In legend of Fig 4C, the authors should indicate whether cells have been treated with Etoposide.

13. The authors should clarify how is defined the mitochondrial aspect ratio (Table S10 and Fig3G).
14. The authors should indicate how long after treatment the images of Fig 5A were taken since Fig 6 demonstrates various results depending on the timing.
15. Figure 6 should be better described in the text.
16. It would be interesting to investigate the effect of etoposide or arsenite treatment on the axonopathy. The authors provide very convincing data after application of PARP1 inhibitor on the proximal segment of iPSC-derived motor neurons (Fig 5E).

Minor concerns:

1. The following typos should be corrected:
 - Line 632 and 633-634: "PARG, DNA-PK, AdOx inhib" should be replaced by "PARG and DNA-PK inhibitors or AdOx".
 - Page 26 of the supplemental word document: replace "PAPR1" by "PARP1".
 - Line 82: "misclocalization" should be replaced by "mislocalization".
2. In the methods, the authors should provide the sequence of the primers used for PCR of the FUS gene after CRISPR-mediated edition.
3. The authors should clarify what is the mutation Y526 in Table S1.

Reviewer #4 (Remarks to the Author):

This manuscript by Naumann and colleagues reports the successful generation of patient-specific iPSC-derived motor neurons (MNs) with FUS-R521C and FUS-P525L mutations. The authors then used microfluidic chamber cultures to show that MNs with FUS-R521C mutation are more prone to "axonopathy", characterized by reduced organelle numbers and loss of mitochondrial potentials and mitochondrial fragmentation in the distal axons. In addition, the authors also showed that MNs with FUS mutations exhibit evidence of defects in DNA damage response, which is dependent on PARP activation. Based on these findings, the authors concluded that DNA damage is upstream of phenotypes induced by FUS mutations.

Overall, this study provides confirmatory data that defects in DNA damage response are early events caused by ALS-related FUS mutations. In this context, the study does not provide additional new insights to what have already reported in the literature. Having said that, the study can be significantly strengthened if the authors can provide rescue experiments and more mechanistic insights on how mutant FUS proteins can dominantly interfere with the DNA damage pathways. In addition to the lack of novelty, the manuscript is very difficult to review because of the profound disorganization of data (e.g. figures cited out of order in many sections) and innumerable grammatical and typographical errors. These problems diffuse the focus and create tremendous confusions to the readers. It will be helpful if the authors seek professional editorial assistance to improve the manuscript. In its current form, this manuscript is best suited for publication in more specialized journals.

Please find below our point-by-point reply:

Reviewer #1 (Remarks to the Author):

General comment:

This is an interesting paper and is well presented. Obvious strengths include the use of isogenic cell lines.

Response: We deeply acknowledge this overall positive review!

Specific comments:

1) This paper focuses on FUS-related neurodegeneration and describes the disruption of the distal axon machinery, DNA repair and the time of cytoplasmic inclusions. However, there does not appear to be an actual read out for cell death/neurodegeneration in their iPSC experiments.

Response: We appreciate this comment and add substantial new data on structural degeneration and cell death. Thereby we show the sequence of dysfunction (axon trafficking) followed by axon degeneration and cell death (completely new Figure 2).

2) The authors use the term distal axonopathy. However, they do not show that the axons are dying back, which would be the sine non-quo of axonopathy.

Response: We agree with the reviewer and add a lot of new experiments (Figure 2, 3, 5). Even though we show a dying back with severe axon degeneration while no neuron loss is seen in vitro (see completely new 2), it is more likely due to a substantial neuronopathy component affecting distal axons than due to a pure axonopathy (see also Reviewer #3). This is supported by the fact, that inducing a distal axonopathy by distal oligomycin A or nocodazole treatment does not induce laser cutting recruitment phenotype or cytoplasmic mislocalization (see Figure5 of the revised manuscript) but whenever FUS recruitment to DNA damage sites is impaired, one also detects a distal axonal phenotype. Thus we rephrased the manuscript accordingly.

3) Furthermore, just because they could identify issues with the distal axon (organelle hang-up), this does not necessarily imply that the problem is in the axon. The axon requires some serious cellular machinery in the cell body for its maintenance. Thus, I think a more likely explanation of their findings is that there is a problem in the cell body and that the consequence of this are then observed in the distal axon. This is especially true given the hypothesis that DNA repair is central to the process of FUS-induced neurodegeneration.

Response: We deeply appreciate this comment. As shown in Response #2, the newly added experimental evidence supports the reviewer's opinion, therefore we rephrased this throughout the manuscript.

Minor points

Page 6: what is meant by "after extended maturation"? Please state the timeframe.

Response: We add this in the revised manuscript (>30 days of maturation)

Page 3: FUS mutations account for less than 1% of sporadic disease, not 3% as stated.

Response: We corrected this mistake.

Reviewer #2 (Remarks to the Author):

General comment:

In this study, Naumann et al. aim to better characterize the initial events in the pathophysiology of FUS-ALS. By generating an isogenic patient iPSC-derived motor neuron model, the authors very elegantly tested the spatiotemporal link between DNA damage, FUS aggregate formation, and neurodegeneration. The major novelty of the study lies in the discovery that FUS-ALS patient mutations cause an increased DNA damage phenotype and distal axonal degeneration prior to FUS aggregation, and that restoring nuclear FUS recruitment to DNA damage sites can modulate the axonal phenotype.

Collectively, this study furthers our understanding of early phenotypes associates with FUS-ALS, providing some interesting insights into targets for therapeutic intervention. Based on these, I think that the manuscript is of strong potential interest for publication in Nature Communications.

Response: We deeply acknowledge this very positive review!

Specific comments:

1) I also feel that some of the connections linking DNA damage to distal axonopathy need to be elaborated to support the model that the authors propose. In particular, while the authors show that treatments, such as inhibiting DNA-PK and PARG inhibition, prevent FUS accumulation in the cytoplasm, DNA-PK inhibition is also likely to inhibit DSB repair through NHEJ. A potential issue with these experiments in the manuscript is that DNA repair is assumed, but not directly measured, following these treatments. Conducting these experiments would help clarify whether it is DNA repair per se, or the cytoplasmic accumulation of FUS, that underlies the development of distal axonopathies.

Response: We thank the reviewer for this comment and add a series of new experiments to get further insights in these mechanisms (revised Fig. 2 & 5 & 6 of the revised manuscript). First we show that these events are specific for interference with DNA damage induction/response signaling and not just secondary to general stress (e.g. OxPhos inhibition by oligomycin A) or axon degeneration by nocodazole. These conditions did affect the axonal trafficking but neither induced impairment of FUS recruitment to DNA damage sites nor cytoplasmic mislocalization of FUS (revised Figure 5&6). Furthermore, while DNA-PK inhibition can rescue etoposide induced phenotypes it cannot rescue FUS recruitment to DDS by PARP1 inhibition clearly showing DNA-PK activation as a downstream mechanisms of DNA damage signaling. While DNA-PK was able to restore FUS mislocalization in controls and FUS mutants also in the presence of PARP1, it couldn't restore FUS recruitment in this co-treatment regimen. This underpins the role of DNA-PK in FUS cytoplasmic export. C-NHEJ and a-NHEJ pathway components were normally expressed in WT and mt FUS MNs (revised Figure 6). FUSmts showed increased DNA-PK phosphorylation, but DNA-PK inhibition reduced DNA-PK phosphorylation as did PARG inhibition suggesting that DDR depends on PARP1 dependent a-NHEJ. FUSmts are able to recover from etoposide induced DNA damage similar to FUSwts but show an increased basal level of DNA damage (revised Fig. 4), however FUSwt showed more intense γ H2AX staining after 1h etoposide treatment (Fig. 4E, F). This fits to data by Wang et al., which showed similar results by FUS knockdown. Thus the main action of FUS-NLS mutants are increased DNA

damage signaling which activates DNA-PK which then enhances the cytoplasmic mislocalization closing a vicious cycle which then causes aggregation formation and neurodegeneration.

Minor Comments:

1) Are there differences in the maturity/electrophysiological properties of the mutant neurons versus control neurons?

Response: Immunostainings of early (TUJ1) and more mature (MAP2) neuronal markers did not reveal any significant difference between disease and control lines. Additional electrophysiological parameters indicative for neuronal maturation (cell capacitance and resting membrane potential) also did not show significant differences between both groups.

2) How do those measurements recorded for the mutant cells (Fig.1G-L) compare to control lines?

Response: As being reported in Naujock et al. (Stem Cells 2016 Jun;34(6):1563-75) we observe a hypoexcitability and hypoactivity phenotype in FUS ALS iPSC derived MNs. We now performed this also with the isogenic FUS-ALS lines and could reproduce this hypoexcitability due to FUS mutations (see new Figure 1M-P of the revised manuscript)

3) What is mutant FUS cell line C10?

Response: This is one clone of the R521L line, we corrected this in the revised manuscript.

4) Figure 1F: in the text it is mentioned that cell death was unaffected, but how was this determined? Quantification of this phenotype would be useful.

Response: This was quantified by appearance of PI staining normalized to Hoechst staining (now added to Figure 1C).

5) It is also very difficult to see the co-localization between methylated-FUS and FUS in the cytosolic aggregates. According to the model presented in the text, methylated-FUS should be primarily cytoplasmic in localization but by immunostaining it looks primarily nuclear?

Response: Methylated FUS should be primarily within the nucleus (see Dormann et al., EMBO 2012, Vol 31, pp4258-75) as shown in our experiments. Higher magnification images and single colour images are now provided in the revised Fig. 1F.

6) Figure 1G: What time point were the electrophysiological recordings conducted at? 30d of differentiation?

Response: All recordings were performed during week 7 of final maturation, as stated in the revised figure legend.

7) "...and showed signs of polyubiquitination (Fig.1D)" is misleading since those experiments were performed in HeLa cells, but in the text it seems to refer to the inclusions observed in iPSC-derived FUS-ALS motor neurons.

Response: We agree with the reviewer and show now new data on iPSC-derived FUS-ALS motor neurons in the revised Fig. 1D.

8) *The order of Fig. 2 and 3 should be changed, as the data presented in Fig. 3 is discussed first in the text.*

Response: We completely re-arranged the figures due to the amount of new data. The original Figure 2 is now Supplemental figure 1. We corrected this in the revised manuscript.

9) *Statistical analysis of data presented in Fig.3D-H, Fig.4E-I, Fig5.F-M is required.*

Response: Please refer to the supplemental file, detailed statistical analysis of all live cell imaging is presented in the supplemental files (Table S2-S16)

10) *Could the authors be clearer about the exact mutant line(s) analyzed throughout the paper? They mention generating iPSCs from 4 different patient lines but it appears as though the majority of data is a collection of data from three different mutant lines (i.e. Fig.3D-H). Were there any phenotypic differences observed between the different mutant lines? A more direct comparison (rather than the data presented separately as per Fig.S4-6) would be helpful.*

Response: On the one hand we agree with the reviewer, however due to the amount of data presented in the manuscript it is difficult to provide this in the main figures but we still prefer to provide these detailed comparisons in the supplemental files. We add additional supplemental files to show these in more detail (Suppl. Fig. S2).

11) *Also, it should be clarified that the FUS corrected data (Fig.3D-H) represents only the R521C corrected line (Fig.2D). How did the authors control for off-target effects of Cas9? How were the two alleles separately sequenced?*

Response: We apologize for causing this confusion and stated this much clearer in the revised manuscript. To check for possible off-target effects of Cas9n both target specific gRNAs (T1 and T2) were checked via the “CRISPR Design” online tool from Zhang laboratory (<http://crispr.mit.edu>) and the “Off-spotter” tool from Pliatsika et al. (Pliatsika, V, and Rigoutsos, I (2015) "Off-Spotter: very fast and exhaustive enumeration of genomic lookalikes for designing CRISPR/Cas guide RNAs" Biol. Direct 10(1):4). The online tools confirmed that there are no genomic off-targets that are targeted from both gRNAs in combination, which could result in the introduction of a DSB at any untargeted site. We included these details in the revised Suppl. Methods.

12) *The phenotypes described in Fig.4 should also be characterized in the isogenic FUS corrected cells.*

Response: We add the respective data (Fig 4E of the revised manuscript)

13) *The finding that DNA DSB levels correlate with mutation/disease severity is very interesting, but the data is not presented in the manuscript other than the correlation values. Could the authors include that data? Additionally, the images corresponding to the data quantified in Fig.4B should be shown i.e. immunostaining for γ H2AX as well as an NPC marker (SOX2, Nestin, etc.).*

Response: We add the respective staining images (revised Fig. 4B). We did not include a figure/table on the correlation on DNA DSB and mutation severity since it is stated in the Nature Communications

author guidelines that unnecessary figures should be avoided: data presented in small tables or histograms, for instance, can generally be stated briefly in the text instead.

14) The finding that etoposide-induced DNA damage caused a distal axon phenotype and not a proximal axonal abnormality (Fig.4) is very interesting. Could the authors comment on this phenomenon? Similarly, could the authors speculate on how restoring FUS recruitment to damage sites rescues the distal axonal phenotype?

Response: We add a lot of new data on structural neurodegeneration and did discuss the reviewer's points in more detail in the revised manuscript (see also responses to Reviewer #1).

15) PARG inhibition resulted in restored recruitment of mutant FUS to damage sites. Since we know that mutant FUS has an impaired interaction with other repair components i.e. HDAC1, it would be interesting to test whether restored recruitment of mutant FUS is sufficient to restore capacity for repair of damage sites via NHEJ.

Response: See also response # 1. We added a lot of new data showing normal expression of both c-NHEJ and a-NHEJ pathway components also in FUSmt MNs (revised Fig. 6). Furthermore, DNA-PK inhibition did not increase cleaved PARP1 expression as a marker for apoptosis. Additionally, DNA-PK inhibition is sufficient to restore FUS mislocalization even in the presence of PARP1 inhibition suggesting DNA damage induction being central in a vicious cycle mainly affecting cytoplasmic shuttling of FUS rather than DNA damage induced apoptosis. FUS mutants are still able to recover from DNA damaging insults (Figure 4 B,C). However, even though really exciting, deciphering the detailed mechanisms how FUS interacts with the different components of c-NHEJ or a-NHEJ or chromatin remodeling is beyond the scope of the current manuscript and clearly a new question to be addressed in a follow up study. Nevertheless we discussed this in more detail in the revised manuscript.

16) The legend for Fig.5A states "mutant P525L FUS failed to repair DNA damage", but the experiment is testing recruitment to sites of DNA damage rather than repair per se. Additionally, the authors should perform immunostaining for a marker of DSBs i.e. γ H2AX to demonstrate the line of recruitment corresponds to actual DNA breaks. Along the same lines, pg.9 "We next asked whether interfering with nucleo-cytoplasmic shuttling of FUS is sufficient to impair DNA damage repair" is an inaccurate statement since capacity of the cells to repair damaged DNA is never measured. Rather, the experiments examine recruitment of FUS to sites of DNA damage.

Response: We totally agree with the reviewer and apologize our imprecise wording. We did rephrase this within the whole manuscript properly.

17) Related to the above, the authors show that inhibition of either DNA-PK or PARG is sufficient to prevent the cytoplasmic accumulation of FUS and ameliorate axonal pathologies. However, DNA-PK activity is also essential for NHEJ. The authors should therefore measure DNA repair following treatment with DNA-PK and PARG inhibitors. These results should further be factored into the proposed model.

Response: We add new data showing that basal DNA damage is significantly higher in FUSmt compared to control, but FUSmt MNs are still able to repair DSBs (see revised Figure 4E-F). DNA-

PK inhibition is able to restore etoposide treatment effects suggesting that DNA damage repair depends on a-NHEJ(revised Fig. 6). Please also refer to comment #15

18) The observation that DNA-PK activity causes the export of FUS following DNA damage generation is potentially interesting, and suggests that in WT MNs, this might serve a physiological function. The authors should at least comment on this, and further clarify based on the experiments suggested in point 17, what happens to DNA repair efficiency when FUS export is inhibited.

Response: We add a lot of additional treatments to further unravel this pathway. As shown in Figure 6, whenever FUS is retained in the nucleus (wtFUS or FUS-NLS mutants), it shows a normal recruitment to DDS except for conditions of upstream inhibition (PARP inhibition). Additionally, we discussed these new results in more detail in the revised manuscript.

19) Pg.13 "...we show here that FUS-ALS is caused by impairment of proper DNA damage response..." This is a bold statement that is not supported by the data presented in the manuscript. What is shown is that the distal axonopathy phenotype is caused by impaired DNA damage response signaling via abnormal FUS targeting.

Response: As mentioned above, we totally agree with the reviewer and rephrased this appropriately throughout the whole manuscript.

20) Pg.14 "...treatment with PARG is sufficient to restore FUS recruitment to DNA damage sites..."

Response: We corrected this typo

Reviewer #3 (Remarks to the Author):

General comment:

Naumann et al. have investigated the relationship between DNA damage response, FUS cytoplasmic aggregation and a distal axonopathy observed in iPS-derived motor neurons carrying ALS-related FUS mutations.....

The link between FUS and the DNA repair machinery has been extensively studied (as appropriately mentioned throughout the manuscript).... Hence, this study builds on several previous reports and confirms the link between FUS mislocalization and DNA damages. The novelty comes mostly from the system used: iPS-derived motor neurons, and isogenic cell lines expressing endogenous levels of wild-type and mutant GFP-FUS. The most important finding is the clear relationship between FUS mutation and abnormal distal axonal trafficking. Importantly, axonal deficit could be restored by targeting FUS nucleocytoplasmic trafficking (AdOX or DNA-PK inhibitor) and enhancing PAR activity (PARG inhibitor) which may have important implication for therapeutic intervention.

Overall, this study clearly demonstrates a central role for DNA damage repair in FUS pathology and the potential of targeting this pathway to restore FUS localization and function. In this reviewer's view the tools and findings from this study are of high relevance for the field.

Response: We are delighted by such a positive review!

Specific comments:

Several concerns mostly related to the presentation of the data should be addressed before consideration for publication in Nature Communication:

1) While the authors propose that DNA damage is an upstream event, the model drawn in Fig6D rather shows that mislocalization preexists and is further enhanced by DNA damages that are less efficiently repaired in presence of FUS mutation. In this respect the title and abstract are misleading and the authors should rather stress the vicious circle enhanced by DNA damages.

Response: We thank the reviewer for this comment. Based on this comment and the additional newly generated data mentioned above (Fig. 2,3, 5,6), we now provide even more experimental evidence that DNA damage is a core element in FUS-ALS but that it mainly enhances FUS mislocalization thereby affecting DNA damage repair thereby inducing a vicious cycle increasing FUS mislocalization. Thus we rephrased this throughout the whole manuscript.

Major concerns:

1) While the introduction of the manuscript is well written, the authors should make additional efforts to describe the results obtained. It is often not clear from the text or legend which lines are being used (especially when figures are labeled only with "FUS" or "Ctrl" (Fig 1C-D and Fig 3)). The authors should clarify whether the graphs on Fig 3D-H represent combined results from all lines.

Response: We apologize for this confusion and clarified this throughout the revised manuscript.

2) Figure 2 is not mentioned in the text, and it would be helpful to have a description of the different lines in the results instead of only in Table S1. Importantly, it is not clear from the text and Fig 2

which lines contain GFP, and whether the R521C line has been corrected to wild-type without GFP (apparently not from Table S1).

Response: There was already a brief description of the lines used in the study, for which we add now more details in the revised manuscript.

3) Similarly, it is not clear which line has been included in Fig 1C (only 2 controls, and mutant “P521L” and “P521C” that are not patient mutations).

Response: We corrected these typos, the mutant lines R521L and R521C were included in this particular analysis.

4) The authors should clarify why Figure 1D has been performed in HeLa cells, what does “ctrl and FUS” mean in this context, and why GFP antibodies recognize multiple bands below 100KD. Most importantly the authors claim that inclusions found in FUS-R521C mutant neurons (Fig 1F) “showed signs of polyubiquitinylation (Fig 1D)” while this has been done in HeLa cells. They need to provide a similar blot from iPSC-derived neurons, and should also co-stain FUS inclusions with ubiquitin antibody.

Response: We are sorry for the confusion; we have now included new data sets and improved western blots using iPSC- derived FUS-ALS MNs (Revised Fig.1D) showing FUS aggregation in dot blot analysis and polyubiquitinylation.

5) The quality of the images is often not appropriate and results difficult to evaluate due to stainings that are too dim (Fig 1B, 1E, 1G, 2B). FUS inclusions are very difficult to see and the authors should provide higher magnification images (Fig 1F and 4E). It is also not possible to determine whether inclusions are labeled with meFUS antibody (they rather appear green than yellow). The authors should present separately all the channels of FUS, meFUS, Hoechst and SMI32 staining at 30 days differentiated motor neurons.

Response: We rearranged the respective figures as mentioned.

6) In legend of Figure 1J, 90.7% of motor neurons fired a single action potential and 32% fired repetitive action potential. These numbers should be clarified.

Response: In the figure legend we stated that at least 90.7% (19 out of 21 cells) fired one single action potential or more. Repetitive action potentials were observed in 33% (7 out of 21 cells) of the MNs.

7) The authors should indicate the differentiation time of the iPSC-derived motor neurons in Figure 3, and determine whether Gamma-H2AX staining is abnormal in iPSC-derived neurons at 14 days of differentiation when they have not observed FUS-R521C mislocalization. It would also be important to clarify whether the time dependency reported in Fig 1 for R521C is also observed with more severe mutations.

Response: We add the new data and details appropriately (Suppl. Fig. 10). The time dependency was similar in the more severe mutations. γ H2AX was already detected after 14days of differentiation prior to cytoplasmic mislocalization underpinning being an early event.

8) Immunohistochemistry in patient samples is used to argue that a distal axonopathy rather than

proximal axonopathy also occurs in vivo. However, images provided in Fig3 I-L are not convincing, with strong spinal motor neuron loss in I and lack of controls in J-K-L preventing the reader to determine whether the thinning of ventral roots is only moderate. In this reviewer's view this claim is neither supported by the results provided nor necessary to the study.

Response: We thank the reviewer for this particularly valuable comment and we regret that the information was not elaborative in Fig3 I-L. In the previous version of the MS, we clearly showed the prominent impairment of distal axons in FUS MNs, and compared our data with autopsy tissue from FUS ALS patients showing MN loss, severe neurogenic atrophy of skeletal muscle together with the degeneration of ventral roots. Based upon reviewer's suggestion we have included new data sets from human postmortem tissue from FUS-ALS patients showing 40-50% loss alpha-MNs in lumbar spinal cord (Fig. 2 G,H) with presence of FUS aggregates in surviving alpha-MNs (Suppl. fig 10C), severe atrophy of skeletal muscles (Supple fig. 10A) and considerable degeneration of the ventral roots (Supple fig 10B) as compared to the controls, all suggesting a substantial neuronopathy component affecting distal axons and the integrity of remaining alpha-MNs. This neuronopathy certainly affects the axons of the integrity of the remaining neurons. So the reviewer is basically right in stating this. Still, there might be effects of pathomechanisms primarily taking place in the distal axons (distal axonopathy in the stricter sense), but we cannot prove this (also not in the additional new in vitro data as discussed with Reviewer #1). This does not change the heart of our story showing that DNA damage is the upstream event but even make the nucleus-to-axon crosstalk clearer. Therefore we rephrased this throughout the whole manuscript.

9) The authors claim at several places that DNA damage influence neurodegeneration but they do not provide any data about neuronal survival. They should replace neurodegeneration by axonopathy (including in the title).

Response: We add new data showing clear signs of structural neurodegeneration and decreased survival as end stage of neurodegeneration (Figure 2 of the revised manuscript) (see also Reviewer #1).

10) The GammaH2AX is Fig 3A, J, K and L is confusing. The images provided in Fig4A after Etoposide treatment of control cells seem to show GammaH2AX staining in the cytoplasm rather than the nucleus. The quality and magnification of the images need to be improved. In addition the authors describe a cytoplasmic staining for GammaH2AX in patient autopsy samples (Fig 4J-L) which would be in contrast with previous reports (Wang 2013, Qiu 2014).

Response: We sincerely regret for the confusion and based upon reviewers suggestions we have included new improved data sets including immunoblot analysis using γ H2AX antibody to detect DSB in these neurons (Revised Fig. 4N,O; Fig. 1D)

Regarding the immunoreactivity of γ H2AX in human autopsy tissue from FUS patients, we consistently observed massive DSB as indicated by a robust immunoreactivity of γ H2AX in the nucleus of α -MNs (Fig.4N, Suppl. Fig. 10D, black arrows) in all the FUS patients included in this study. However, in addition few α -MNs of lumbar spinal cord from one particular patient showed peculiar γ H2AX labelling both in the nucleus (Suppl. Fig. 10 D; black arrows) as well as in the cytoplasm (Suppl. Fig. 10 D ;white arrows). This particular patient was with FUS Y526C mutation, severely affected and died at the age of 35. This mutation is already known to be associated with aggressive juvenile ALS (PMID: 28054830). Autopsy tissue from this patient showed massive α -MNs

loss (30-35%), with severe skeletal muscle atrophy and only very few surviving neurons in other spinal cord levels. More than 80% of the remaining α -MNs of lumbar spinal cord showed abnormal morphology and FUS aggregation accompanied by abnormal nuclear morphology. Thus based upon these observations γ H2AX labelling in the cytoplasm co-localized with FUS aggregation might be pathologically relevant. Future work beyond the scope of this MS is required to understand the pathological consequences of this particular FUS Y526C mutation on DDR. Interestingly however this pattern of staining was not observed in any other FUS patients (with R521C mutation) and also in any sporadic ALS cases (data not shown). We performed co-immunolabeling together with FUS antibody and confirmed the aberrant co-localization of γ H2AX accumulations with FUS aggregates in the cytoplasm of surviving α -MNs (Fig.4L: yellow arrows) in the lumbar spinal cord of this particular patient. Taken together, our results suggest the notion that DNA damage is probably an early event in FUS pathophysiology. We have included all the above data together with the respective explanation in the revised version of the MS.

11) The authors should discuss the difference between their experiments showing that mutant GFP-P525L-FUS is not recruited to the DNA damage site (DDS) and results from Wang et al. (2013), Rulten et al. (2015) and Mastrocola et al. (2013) where overexpressed mutant R521C, R521G, or R524S-FUS are recruited to DDS.

Response: While Rulten et al also showed relevant reduction of FUS recruitment, this was by far not that obvious in the report by Mastrocola et al. and was not seen in the Wang paper. Differences most likely arose from different techniques of expressing the FUS protein (only in our study by mutating the endogenous protein) and the cell types used (only we used patient-derived human MNs). We include a detailed discussion on this in the revised manuscript.

12. In legend of Fig 4C, the authors should indicate whether cells have been treated with Etoposide.

Response: We clarified this in the revised manuscript

13. The authors should clarify how is defined the mitochondrial aspect ratio (Table S10 and Fig3G).

Response: The aspect ratio refers to the eclipse fitted to mitochondria. Each eclipse is defined by its long and short radius, the aspect ratio is simply the long divided by the short radius and serves as a measure for mitochondria elongation. The analysis of the aspect ratio is described in Supplemental Methods, section 'Static analysis of cell' organelles.

14. The authors should indicate how long after treatment the images of Fig 5A were taken since Fig 6 demonstrates various results depending on the timing.

Response: We assume the reviewer is referring to Fig 5C and 5K, as 5A shows only untreated cells. As for 5C and 5K, cells were treated for 24 hrs before imaging. We have now specified this information with better clarity in Supplemental Methods, section 'Treatment/Inhibitors' as well as in the figure legend.

15. Figure 6 should be better described in the text.

Response: We focus more on the vicious cycle and thus do more explain the Figure 6.

16. It would be interesting to investigate the effect of etoposide or arsenite treatment on the axonopathy. The authors provide very convincing data after application of PARP1 inhibitor on the proximal segment of iPS-derived motor neurons (Fig 5E).

Response: We include these data in the revised manuscript (see Fig. 4H and Suppl. Movies??).

Minor concerns:

1) The following typos should be corrected:

- Line 632 and 633-634: “PARG, DNA-PK, AdOx inhib” should be replaced by “PARG and DNA-PK inhibitors or AdOx”.

- Page 26 of the supplemental word document: replace “PAPRI” by “PARP1”.

- Line 82: “misclocalization” should be replaced by “mislocalization”.

Response: We corrected these typos

2. In the methods, the authors should provide the sequence of the primers used for PCR of the FUS gene after CRISPR-mediated edition.

Response: We add this in the revised manuscript.

3. The authors should clarify what is the mutation Y526 in Table S1.

Response: Y526C, we add this information and the paper describing this mutation in the revised manuscript.

Reviewer #4 (Remarks to the Author):

General comments:

Overall, this study provides confirmatory data that defects in DNA damage response are early events caused by ALS-related FUS mutations. In this context, the study does not provide additional new insights to what have already reported in the literature.

Response: We disagree with this reviewer's overall statement as do the other three reviewers as well. As reviewer # 3 stated in detail, we build up on preexisting data on FUS and DNA damage providing significant novel insights in how FUS mutations and impaired DNA damage sensing leads to neurodegeneration and aggregation formation by inducing a vicious cycle which is mainly driven by the impaired DNA damage response signaling. We further provide new data on the role of FUS and respective mutants in NHEJ pathways. We for the first time characterize the unexpected role of DNA-PK in FUS-ALS and showed, that FUSmt phenotypes could be restored by targeting FUS nucleocytoplasmic trafficking (AdOX or DNA-PK inhibitor) and enhancing PAR activity (PARG inhibitor) which may have important implication for therapeutic intervention.

Specific comments:

1) The study can be significantly strengthened if the authors can provide rescue experiments and more mechanistic insights on how mutant FUS proteins can dominantly interfere with the DNA damage pathways.

Response: We added (to the plethora of already shown rescue experiments) new mechanistic data on neurodegeneration (Fig. 2), DNA damage sensing (Fig. 5 & 6) and the involved DNA damage repair pathway (Fig. 6) adding additional novelty to the revised manuscript.

2) The manuscript is very difficult to review because of the profound disorganization of data (e.g. figures cited out of order in many sections) and innumerable grammatical and typographical errors. These problems diffuse the focus and create tremendous confusions to the readers. It will be helpful if the authors seek professional editorial assistance to improve the manuscript.

Response: We deeply apologize and completely re-arranged the manuscript. The revised manuscript was edited by a native speaker (Tanya Levin).

Reviewer #1 (Remarks to the Author):

The authors have responded to the suggestions raised by myself and the other reviewers. The manuscript is improved and reads very well indeed. No further suggestions.

Reviewer #2 (Remarks to the Author):

In the revised manuscript, the authors have provided new evidence supporting that DNA damage is the initial event prior to FUS aggregation and axonal degeneration in FUS-ALS. It is very intriguing that inducing DNA damage triggers cytoplasmic localization of wild-type FUS, which mimics the phenotypes of FUS-NLSmt and further strengthens the importance of maintaining genome integrity in the brain. The figures are greatly improved, and the authors have addressed several of the concerns raised previously. Below you will find a few comments that will be necessary to address prior to publication in Nature Communications.

(1) In Figure 5, the current evidence only supports that the impairment of nucleo-cytoplasmic FUS shuttling leads to the failure of recruitment to break sites but not increased DNA damage. Along the same lines, the legend for Figure 5A "Mutant P525L FUS failed to repair DNA damage" is an inaccurate statement. It will be more appropriate if the authors perform γ H2AX staining as I suggested previously or rephrase the section to not over interpret the data.

(2) There are some typos or incorrect information presented in the figure or text as below:

a. Page 21, line 458: Western blot analysis (FUS/H2Ax) were performed as described in Prause et al. -> The authors did not perform any γ H2AX western blotting, do the authors mean immunostaining?

b. Page 21, line 470: phospho-DNA-PK T2609 (ab18356), LIG1 (ab177946), KU70 (ab31114), GFP (ab290, Abcam). -> Data of phospho-DNA-PK T2609 is not presented in the manuscript.

c. Page 36, Figure 4G, BP53BP1 -> 53BP1

d. Page 36, line 745: Fig.4: DDR signaling is involve in FUS-NLS pathophysiology -> DDR signaling is involved in FUS-NLS pathophysiology

e. Page 42, line 812: how DNA-PK inhib. or AdOx drove FUS-GFP back into the nucleus under conditons that -> conditions

f. Movie S4, Etoposid ->Etoposide

Reviewer #3 (Remarks to the Author):

In this revised manuscript Naumann et al. have addressed several concerns raised by the reviewers and included additional information supporting an upstream role of PAR activity in FUS recruitment to DNA breaks.

The study shows a striking distal axonopathy in FUS mutant neurons and provides evidence for a central role for DNA damage repair in FUS pathology and the potential of targeting this pathway to restore FUS localization and function. While this study is of high interest for the field, several issues were not addressed in this revised manuscript.

1. There is a remaining confusion as to whether FUS mutant iPS-derived neurons display increased neurodegeneration. Indeed, in the title and throughout the text the authors claim a role for DNA repair defects in FUS-related neurodegeneration. In the response to reviewers they claim that

Figure 2 shows increased cell death. However, results in Fig 2F and Fig 2J are contradictory with Figure 2F showing a normal % NeuN positive nuclei at 21 DIV and 60 DIV, while Figure 2I and J show a decreased survival. The cell line (specific FUS mutation), the time point and the assay used in Fig 2I-J are not described. Survival defect (if present) may be quite different depending on the mutation and time point studied. The text is also extremely confusing:

- Line 138: "neither increased cell death nor pathological aggregation in the early stages of differentiation (Fig 1C)".
- Line 160: "there was still no relevant neuron loss.... Nevertheless there was a decrease in MN survival in FUSmt"
- Line 400: "and finally motor neuron cell death (Fig 2,3)" (Fig 3 has nothing related with this issue).
- Line 402: "only marginal cell death at time points of severe axon degeneration (Fig 2)".

The authors need to clarify this point with a robust, well described, assay for neurodegeneration, and a systematic study of all the cell lines.

2. The stainings provided for GammaH2AX in Figure 4, and even more so in Fig S10, are not convincing (images are too small, dim, arrows are pointing to some staining while other fluorescent regions are ignored), and no control is provided for Fig S10D-E. There is also quite some confusion about the relationship between specific mutations and the extent of GammaH2AX staining. The authors claim that "the amount of DNA DSBs correlated with the severity of the underlying mutation and the disease onset in the patients ($r=-0.785$; $p=0.012$, pearson correlation)", but it is not clear how they obtained this. Indeed, the authors do not provide quantification of gammaH2AX for each mutation (from the text it seems that #DSBs is quantified in the P525L line in Fig 4C-D). In addition, there are only 3 patients with age of onset at 57, 61, and "not determined but in the twenties".

3. In line with the two previous concerns, the last sentence of the manuscript appears inadequate: "our study presents a correlation between mutation severity, age of onset, amount of DNA DSBs, appearance of neurodegenerative phenotypes....".

4. The proposed increased polyubiquitination in mutant FUS line is not convincing in Figure 1D since the loading is not equivalent between control and mutant (see Tubulin).

5. The presentation of the results remains to be improved with very small images and graphs and often unreadable labeling. The order of the Figures is often not respected in the text. Finally, labeling and legends of the figures is still imprecise in particular concerning information about the line/mutation investigated ("FUS" versus "Ctrl").

Reviewer #4 (Remarks to the Author):

The revised manuscript by Naumann and colleagues has included many additional data to strengthen its conclusion. In addition, the authors made a strong effort to improve the readability and the overall organization of their manuscript. There are a few areas that require additional attention in order to support the conclusion and provide clarity and accuracy.

1. The data in the newly added Figure 6G require quantification to support the descriptions in lines 321 to 329. This is very important because the new statement regarding the involvement of FUS and FUS mutants in NHEJ.

2. It is this reviewer's opinion that the schematic diagram in Figure 6H is more suitable for a review article as the current data in this manuscript do not fully address the cause-effect and the proposed "vicious" cycle.

3. In the 2nd paragraph of the Discussion, the authors cited three references (12-14), and the stated that "These differences most likely arise from different cell types (.....) and the technique of FUS expression (overexpression compared to endogenous expression levels, the latter used for the first time in the current study)(lines 345 to 348). Although this statement is factually correct, it does not make logical sense because the expression levels of wild type and mutant FUS in all the U2OS lines reported by Wang et al (ref #14) were carefully controlled in all reporter lines. It is most likely that the differences in FUS recruitment to DNA damage foci in the current study are due to cell type specific effects.

4. Finally, in Abstract the authors stated, "Our work challenges the current thinking about ALS by providing evidence that the key pathophysiologic event is upstream of aggregate formation." This reviewer believes this statement needs to be revised to make more sense from both logical and scientific standpoints. What exactly is the "current thinking" that was "challenged" by this study? Is it about the role of FUS in "DNA damage" as the Title suggested? Or, is it about "distal axonopathy"? The distal axonopathy phenotype is certainly novel, however, this study does not provide mechanistic insights on how FALS mutations in FUS contribute to this phenotype, and whether the distal axonopathy is connected to DNA damage at all. It is important to note that other reviewers (e.g. Reviewer #3, Point 1) shared the same opinion about the title and abstract, calling it "misleading". But this critique was not fully addressed in the revised Abstract and other areas in the revised manuscript.

Reviewer #1 (Remarks to the Author):

General comment:

The authors have responded to the suggestions raised by myself and the other reviewers. The manuscript is improved and reads very well indeed. No further suggestions.

Response: We are very grateful for this positive response. We appreciate that this reviewer also stated that we responded to the comments raised by the other reviewers as well.

Reviewer #2 (Remarks to the Author):

General comment:

In the revised manuscript, the authors have provided new evidence supporting that DNA damage is the initial event prior to FUS aggregation and axonal degeneration in FUS-ALS. It is very intriguing that inducing DNA damage triggers cytoplasmic localization of wild-type FUS, which mimics the phenotypes of FUS-NLSmt and further strengthens the importance of maintaining genome integrity in the brain. The figures are greatly improved, and the authors have addressed several of the concerns raised previously. Below you will find a few comments that will be necessary to address prior to publication in Nature Communications.

Response: We deeply acknowledge this very positive review and address all minor concerns mentioned below!

Specific comments:

1) In Figure 5, the current evidence only supports that the impairment of nucleo-cytoplasmic FUS shuttling leads to the failure of recruitment to break sites but not increased DNA damage. Along the same lines, the legend for Figure 5A “Mutant P525L FUS failed to repair DNA damage” is an inaccurate statement. It will be more appropriate if the authors perform γ H2AX staining as I suggested previously or rephrase the section to not over interpret the data.

Response: We agree with the reviewer. Indeed we show a failure of FUS recruitment but not increased DNA damage in Figure 5. We revised our wordings in the figure legend accordingly.

*2) There are some typos or incorrect information presented in the figure or text as below:
a. Page 21, line 458: Western blot analysis (FUS/H2Ax) were performed as described in Prause et al. -> The authors did not perform any γ H2AX western blotting, do the authors mean immunostaining?*

Response: We regret the confusion. We performed FUS western blots (see Figure 1D), but not for γ H2AX, and we show immunostaining for both. We revised this section accordingly.

b. Page 21, line 470: phospho-DNA-PK T2609 (ab18356), LIG1 (ab177946), KU70 (ab3114), GFP (ab290, Abcam). -> Data of phospho-DNA-PK T2609 is not presented in the manuscript.

Response: We regret for wrongly citing the used antibodies: The phospho-DNA-PK S2056 (ab18192, Abcam) was used. We corrected this in the revised manuscript.

c. Page 36, Figure 4G, BP53BP1 -> 53BP1

Response: Corrected.

d. Page 36, line 745: Fig.4: DDR signaling is involve in FUS-NLS pathophysiology -> DDR signaling is involved in FUS-NLS pathophysiology

Response: Corrected.

e. Page 42, line 812: how DNA-PK inhib. or AdOx drove FUS-GFP back into the nucleus under conditons that -> conditions

Response: Corrected.

f. Movie S4, Etoposid ->Etoposide

Response: Corrected.

Reviewer #3 (Remarks to the Author):

General comment:

In this revised manuscript Naumann et al. have addressed several concerns raised by the reviewers and included additional information supporting an upstream role of PAR activity in FUS recruitment to DNA breaks.

The study shows a striking distal axonopathy in FUS mutant neurons and provides evidence for a central role for DNA damage repair in FUS pathology and the potential of targeting this pathway to restore FUS localization and function. While this study is of high interest for the field, several issues were not addressed in this revised manuscript.

Response: We appreciate this overall positive review and addressed all remaining concerns as described below.

Specific comments:

1.) There is a remaining confusion as to whether FUS mutant iPS-derived neurons display increased neurodegeneration. Indeed, in the title and throughout the text the authors claim a role for DNA repair defects in FUS-related neurodegeneration. In the response to reviewers they claim that Figure 2 shows increased cell death. However, results in Fig 2F and Fig 2J are contradictory with Figure 2F showing a normal % NeuN positive nuclei at 21 DIV and 60 DIV, while Figure 2I and J show a decreased survival. The cell line (specific FUS mutation), the time point and the assay used in Fig 2I-J are not described. Survival defect (if present) may be quite different depending on the mutation and time point studied. The text is also extremely confusing:

- Line 138: “neither increased cell death nor pathological aggregation in the early stages of differentiation (Fig 1C)”.

- Line 160: “there was still no relevant neuron loss.... Nevertheless there was a decrease in MN survival in FUSmt”

- Line 400: “and finally motor neuron cell death (Fig 2,3)” (Fig 3 has nothing related with this issue).

- Line 402: “only marginal cell death at time points of severe axon degeneration (Fig 2)”.

The authors need to clarify this point with a robust, well described, assay for neurodegeneration, and a systematic study of all the cell lines.

Response: We thank the reviewer for this particularly valuable comment and, as he mentioned, we found prominent degeneration of the axons but rather mild signs of neuronal cell body degeneration at DIV 60. We add now a series of new experiments including 110 days of differentiation/maturation as well as caspase3 stainings at different time-points (see revised Fig. 2F-K, Fig. S11, page 8 of the revised manuscript). By doing so, we show already increased caspase3 positive MNs in FUSmts compared to controls at DIV 60 which was even higher at DIV 110. At DIV110 this also results in reduced counts for neurons and motor neurons. With these data in hand we clearly show the time course of neurodegeneration starting with dysfunctional axons, followed by structural degeneration of the axons and finally the death of the entire MNs. This assay per se is - to the best of our knowledge - shown for the first time *in vitro* using hiPSC models.

Of note, many authors described distal axonopathy in ALS including sporadic ALS (see, for example, Fischer et al, *Exp Neurol.* 2004 Feb;185(2):232-40; Dadon-Nachum et al., *J Mol Neurosci* (2011) 43:470–477; Frey et al., *The Journal of Neuroscience*, April 1, 2000, 20(7):2534–2542; Fischer et al., *Neurodegenerative Dis* 2007;4:431–442) with the typical sequence of neuromuscular denervation followed by motor nerve thinning and finally anterior horn cell loss. Thus, our *in vitro* model mimics these *in vivo* data.

2.) *The stainings provided for GammaH2AX in Figure 4, and even more so in Fig S10, are not convincing (images are too small, dim, arrows are pointing to some staining while other fluorescent regions are ignored), and no control is provided for Fig S10.*

Response: We agree with the reviewer and now provide higher magnification/resolution images in the revised versions and add a control in revised Fig. S10.

There is also quite some confusion about the relationship between specific mutations and the extent of GammaH2AX staining. The authors claim that “the amount of DNA DSBs correlated with the severity of the underlying mutation and the disease onset in the patients ($r=-0.785$; $p=0.012$, pearson correlation)”, but it is not clear how they obtained this. Indeed, the authors do not provide quantification of gammaH2AX for each mutation (from the text it seems that #DSBs is quantified in the P525L line in Fig 4C-D). In addition, there are only 3 patients with age of onset at 57, 61, and “not determined but in the twenties”.

Response: We do agree with the reviewer that we might overstate this correlation having only samples from 3 patients. Therefore we removed this section to avoid misinterpretation.

3.) *In line with the two previous concerns, the last sentence of the manuscript appears inadequate: “our study presents a correlation between mutation severity, age of onset, amount of DNA DSBs, appearance of neurodegenerative phenotypes....”.*

Response: We have also rephrased this section. (also see above)

4.) *The proposed increased polyubiquitination in mutant FUS line is not convincing in Figure 1D since the loading is not equivalent between control and mutant (see Tubulin).*

Response: We regret the confusion. Addressing the reviewer’s comment we have now replaced the previous data with more convincing WB images showing increased polyubiquitination and FUS aggregation in mutant FUS cells (revised Fig.1D).

5) *The presentation of the results remains to be improved with very small images and graphs and often unreadable labeling. The order of the Figures is often not respected in the text. Finally, labeling and legends of the figures is still imprecise in particular concerning information about the line/mutation investigated (“FUS” versus “Ctrl”).*

Response: We truly regret the errors. We now thoroughly revised the entire manuscript, and also rearranged the respective figures and legends.

Reviewer #4 (Remarks to the Author):

General comments:

The revised manuscript by Naumann and colleagues has included many additional data to strengthen its conclusion. In addition, the authors made a strong effort to improve the readability and the overall organization of their manuscript. There are a few areas that require additional attention in order to support the conclusion and provide clarity and accuracy.

Response: We are delighted by the overall positive review and addressed all remaining minor concerns (see below).

Specific comments:

1) *The data in the newly added Figure 6G require quantification to support the descriptions in lines 321 to 329. This is very important because the new statement regarding the involvement of FUS and FUS mutants in NHEJ.*

Response: We appreciate the reviewer’s concern and apologize for not having done the quantification in the previous version, which is now included in the revised Figure 6G-I of the revised manuscript showing clear results.

2. *It is this reviewer’s opinion that the schematic diagram in Figure 6H is more suitable for a review article as the current data in this manuscript do not fully address the cause-effect and the proposed “vicious” cycle.*

Response: Figure removed.

3.) *In the 2nd paragraph of the Discussion, the authors cited three references (12-14), and the stated that “These differences most likely arise from different cell types (.....) and the technique of FUS expression (overexpression compared to endogenous expression levels, the latter used for the first time in the current study)(lines 345 to 348). Although this statement is factually correct, it does not make logical sense because the expression levels of wild type and mutant FUS in all the U2OS lines reported by Wang et al (ref #14) were carefully controlled in all reporter lines. It is most likely that the differences in FUS recruitment to DNA damage foci in the current study are due to cell type specific effects.*

Response: We do agree that the expression levels were carefully controlled in the Wang et al study and re-phrased this paragraph appropriately. Nevertheless, we also mention that differences might also arise due to technical issues concerning ectopic vs. endogenous expression (see revised paragraph 2 of the discussion section).

Reviewers' Comments:

Reviewer #3:

Remarks to the Author:

In this third version of the manuscript the authors have answered to the main comments of the reviewers. In this referee's view, this work will be of high interest to the field and is appropriate for publication in Nature Communications.

Reviewer #4:

Remarks to the Author:

The revised manuscript has addressed my previous concerns and therefore is much improved compared to prior version.